# A Reformatory Model Incorporating PNGV Battery and Three-Terminal-Switch Models to Design and Implement Feedback Compensations of LiFePO$_4$ Battery Chargers

Kai-Jun Pai 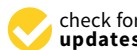

Department of Electrical Engineering, Ming Chi University of Technology, New Taipei City 24301, Taiwan; carypai@mail.mcut.edu.tw; Tel.: +88-622-908-9899 (ext. 4849)

**Abstract:** This study developed and implemented a LiFePO$_4$ battery pack (LBP) rapid charger. Using the three-terminal switch and partnership for a new generation of vehicles (PNGV) battery models, this study could obtain a small-signal system matrix to derive transfer functions and further analyze frequency responses for the charge voltage and current loops; therefore, both voltage and current feedback controllers could be designed to fulfill the constant-voltage (CV) and constant-current (CC) charges. To address practical applications, the proposed equivalent model also considered the wire resistance-inductance of the power cable. According to the derived high-order transfer function, the pole-zero break frequency in the Bode plot was observed that approximated the practical measurement; therefore, the pole-zero compensation could be accomplished for both charge loop requirements. Moreover, the design features for implementing the CV and CC charges are presented in detail herein, and the current overshoot during the start-up phase could be mitigated using the method of zero break frequency shifting and a novel proportional shifting proportional-integral control. The LBP parameter estimations, model construction processes, and frequency response analyses are also presented. The feedback compensation design based on the proposed model was validated through simulations and experiments. The results were determined to be in excellent agreement with theoretical derivations.

**Keywords:** rapid charger; PNGV battery model; three-terminal switch model; small-signal; proportional shifting proportional-integral control

## 1. Introduction

The greenhouse effect, resulting in climate change, has become a major problem, and ecofriendly technologies for producing clean energy are paramount to alleviating greenhouse gas emissions. Commonly-used rechargeable batteries including lead-acid, Ni-Cd, Ni-MH, and Li-ion batteries have been applied in smartphones, laptops, electric screwdrivers, and other portable instruments as well as in electric forklifts and other electric vehicles. Compared with other secondary batteries, Li-ion batteries have the highest power and energy density [1]; therefore, Li-ion cells inside a battery pack are a suitable choice for electric vehicles [1–3].

The Lithium iron phosphate (LiFePO$_4$) is suitable for the positive electrode material in batteries because the strong P–O covalent bonds in the LiFePO$_4$ lattices do not decompose easily. The LiFePO$_4$ battery possesses excellent thermal stability; even under high-temperature or overcharge conditions, they rarely overheat because the LiFePO$_4$ lattice does not easily collapse and oxidize. Moreover, LiFePO$_4$ batteries are suitable for powering electric vehicles because they offer multicycle

charge/discharge and nontoxicity [4–6]. However, the energy density of LiFePO$_4$ battery is lower than LiMn$_2$O$_4$ and LiCoO$_2$ batteries.

The constant-voltage (CV) and constant-current (CC) outputs of rapid chargers are necessary functions for the charge of large-capacity LiFePO$_4$ battery pack (LBP). To accomplish CV and CC charges, the voltage feedback controller (VFC) and current feedback controller (CFC) must be designed based on the frequency responses of charge loops. The battery electrical model is a critical role because a suitable model can present a practical frequency response. The suitable electrical model for a Li-ion battery include the R$_{int}$ model, the resistance–capacitance (RC) model, the Thevenin model, the modified Thevenin model, and the Partnership for a New Generation of Vehicles (PNGV) battery model [7–9]. The PNGV battery model [10–12] was adopted in this study because its parameters can be estimated using the pulse-current charge method to obtain the battery voltage–time characteristic, and the formula for the electric charge can be adopted to calculate the model parameter [10].

In this study, the developed LBP rapid charger (LBPRC) comprised a safety-standard circuit, a power factor corrector, and a DC–DC converter. The topology of the DC–DC converter was a phase-shifted full-bridge with parallel current-doubler rectification (PSFB-PCDR) incorporating the VFC and CFC to achieve the CV or CC output mode. In addition, a three-terminal switch (TTS) model was applied to simplify the PSFB-PCDR [13–15]. Moreover, studies [16–19] have mentioned several charge strategies for rechargeable batteries. In [16], the power stage topology of the battery charger was a boost converter, which could step up the low input voltage from the photovoltaic cell, and using the control technologies of the maximum power point tracking and the pulse-charge scheme, the fast maximum power point tracking could be achieved during a narrow charge period. In [17], the battery charge combined the bridgeless power factor correction (PFC) with the PSFB converter; this power topology design could easily achieve the series or parallel combination of battery charger for electric vehicle applications. Moreover, the CC–CC–CV charge strategy was applied; this charge method should be effective to extend the LiFePO$_4$ battery lifespan. In [18], a two-switch buck converter was used as the power stage topology, which could be applied in an electrically controlled pneumatic brake system; the CC–CV charge strategy was implemented to charge the LiFePO$_4$ battery. In [19], the series resonant converter with the synchronous rectification was the power stage topology of battery charger; the charge strategy adopted the CC–CV method. From [16–19], comparisons of power converter topologies and charge strategies are listed in Table 1.

**Table 1.** Comparisons of power converter topologies and charge strategies.

| Reference | [16] | [17] | [18] | [19] | This Work |
|---|---|---|---|---|---|
| Application | Photovoltaic system | Electric vehicle | Railway | Not mentioned | Electric vehicle |
| Power stage topology | Boost | Bridgeless PFC and PSFB converter | Buck | Series resonant converter and synchronous rectification | PFC and PSFB-PCDR |
| Battery | Lead-acid | LiFePO$_4$ | LiFePO$_4$ | LiFePO$_4$ | LiFePO$_4$ |
| Electrical model of battery | Not mentioned | Not mentioned | Not mentioned | Not mentioned | PNGV battery model |
| Charge strategy | CC–CV–pulse | CC–CC–CV | CC–CV | CC–CV | CC–CV |
| VFC design | Not mentioned | Not mentioned | Not mentioned | Not mentioned | Proportional-integral (PI) control |
| CFC design | Not mentioned | Not mentioned | Not mentioned | Not mentioned | PI control and novel PSPI control |

Some previous studies [20–22] adopted the simple R$_{int}$ model or the RC battery model to establish the system small-signal model; however, this simple, one-order battery model cannot reflect the practical frequency response and the break frequency, and therefore, the pole-zero compensation and the dynamic characteristic improvement could not be performed precisely. In this study, the high-order

transfer function (TF) was established to ameliorate the deficiencies of the previous studies [20–22], and the scheme of current overshoot mitigation using a new proportional shifting proportional-integral (PSPI) control can be achieved. Table 2 presents a comparison of the developed method in the present study with other charger technologies and control methods.

**Table 2.** Comparisons of charger technologies and control methods.

| Reference | [20] | [21] | [22] | This Work |
|---|---|---|---|---|
| Battery type (cell or pack) | Pack | Single cell | Single cell | Pack |
| Battery electrical model | Simple ($R_{int}$ model) | Simple (RC model) | Simple (RC model) | Complex (PNGV battery model) |
| TF order of charger incorporating battery | 2 | 2 | 2 | 5 |
| Small-signal analysis for CV charge | Brief survey | Brief survey | Brief survey | Detailed explanation |
| Small-signal analysis for CC charge | Brief survey | Brief survey | Brief survey | Detailed explanation |
| Bode plot | Simulation | Simulation | Simulation | Simulation and practical measurement |
| Voltage-loop compensation | Simple design | Simple design | Simple design | Complete design |
| Current-loop compensation | Simple design | Simple design | Simple design | Complete design |
| Mitigation of start-up current overshoot | Not mentioned | Not mentioned | Not mentioned | Novel PSPI control |

This paper is divided into seven sections. Section 2 discusses the PNGV battery model with parameter estimation. In Section 3, the PSFB-PCDR employs the TTS model incorporating PNGV battery model to establish the system matrix. Frequency response simulations are provided in Section 4. The CV and CC feedback controllers designed on the basis of these simulations are presented in Section 5, and measurements and experimental results are presented in Section 6. The concluding remarks and primary contributions of this study are given in Section 7.

## 2. Estimating PNGV Battery Model Parameters

A PNGV battery model representing the LBP is depicted in Figure 1. The model includes a polarization capacity capacitance $C_t$, a battery capacity capacitance $C_x$, a polarization internal resistance $R_t$, an ohmic internal resistance $R_{oir}$, and an open-circuit voltage $v_{ocv}$ [8,10,12,23,24].

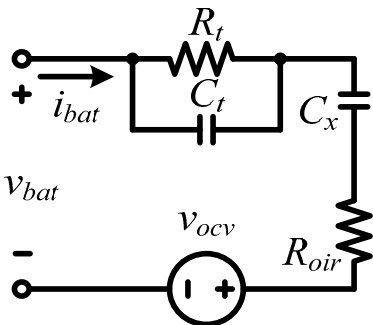

**Figure 1.** PNGV battery model.

Figure 2a illustrates the system measurement configuration for estimating the parameter of the PNGV battery model. The LBPRC inlet was an AC source input, and the positive/negative electrode of the LBP was connected to the LBPRC outlet. The voltage-date collector GL240 (Graphtec Corp., Totsuka-ku, Yokohama, Japan) can be used to record the LBP voltages every 10 ms (the minimum sampling time of the GL240) and transmit them to the computer via USB; therefore, the LBP adopting a pulse-current charge method can obtain the voltage–time characteristics, as depicted in Figure 2b.

Both results produced by the charge equation and the Ohm's law can be used to calculate the parameter of the PNGV battery model.

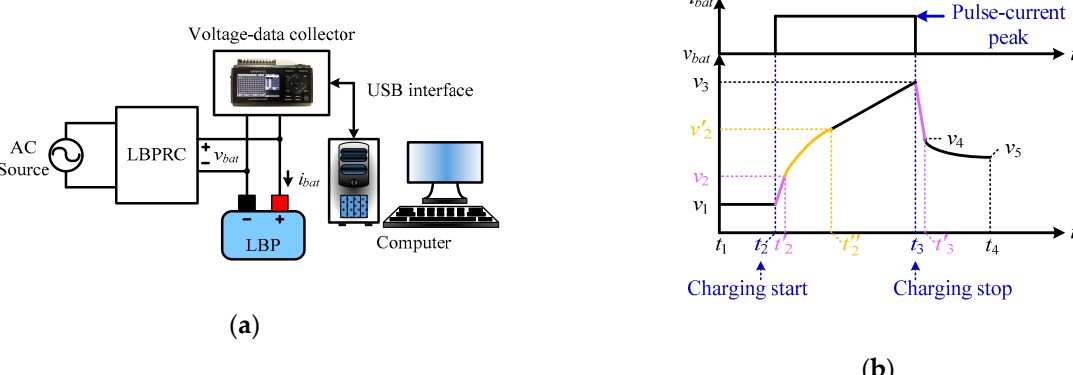

**Figure 2.** Parameter measurement of PNGV battery model. (**a**) Measurement configuration diagram. (**b**) Voltage−time characteristic using pulse-current charge.

## 2.1. Battery Capacity Capacitance

In this study, $C_x$ represents the LBP capacity that can be expressed as $C_x = i_{bat} \times \Delta t / \Delta v_{bat}$ [10], this equation is from the charge $\Delta Q = C_x \times \Delta v_{bat} = i_{bat} \times \Delta t$; the $\Delta v_{bat}$ is the LBP voltage variation with the time interval $\Delta t$; the $i_{bat}$ is the pulse-current peak. During the time interval $[t_2{:}t_3]$, the LBP is charged by a pulse-current, hence the $\Delta t$ equals to $t_3 - t_2$. Moreover, because of the LBP is affected by the tardy ion diffusion process, the LBP open-loop voltage needs an emancipated time to recovery after the charge stop, and therefore, the $\Delta v_{bat}$ equals to the LBP's voltage difference $v_5 - v_1$ (Figure 2b). As a result, the $C_x$ can be rewritten as follows:

$$C_x = i_{bat}\Delta t / \Delta v_{bat} = i_{bat}(t_3 - t_2)/(v_5 - v_1) \tag{1}$$

It is noticeable that the LBP voltage from $v_3$ to $v_4$ has a time delay from $t_3$ to $t_3'$, because the minimum sampling time of the GL240 should be considered. However, this time delay does not affect the $C_x$ estimation.

## 2.2. Ohmic Internal Resistance

When the Li-ion battery is charged or discharged, migratory electrons pass through metallic elements and chemical materials; these substances inside the Li-ion batteries are similar to a resistance obstructing the electron movement, hence the PNGB battery model using the $R_{oir}$ (Figure 1) represents the LBP internal resistance. Moreover, the phenomenon of ohmic voltage drops occurring at the charge start (at $t_2$) and stop (at $t_3$) times are caused by the $R_{oir}$; at $t_2$ and $t_3$, the $R_{oir}$ can be expressed as the $R_{oir(st)}$ and $R_{oir(sp)}$, respectively. According to the literature [10] and Ohm's law, the $R_{oir(st)}$ and $R_{oir(sp)}$ are respectively expressed as follows:

$$R_{oir(\text{st})} = (v_2 - v_1)/i_{bat} \tag{2}$$

$$R_{oir(\text{sp})} = (v_3 - v_4)/i_{bat} \tag{3}$$

where $v_2 - v_1$ and $v_3 - v_4$ represent the LBP voltage differences. From practical measurement and estimation, $R_{oir(st)}$ can approximate to $R_{oir(sp)}$. To simplify this study derivations, the equation $[R_{oir(st)} + R_{oir(sp)}]/2$ can be used to obtain an average $R_{oir}$ for further analyses and designs.

### 2.3. Polarization Capacity Capacitance and Internal Resistance

Because of the charge transfer and diffusion of electrochemical reaction inside the battery, the $v_2$ exponentially changes to $v'_2$ during the time interval $[t'_2 : t''_2]$. Moreover, the resistance–capacitance time constant $\tau$ can be obtained by the $R_t \times C_t$, this rule can refer to the literature [7,10,25]. From [10], $C_t$ can be expressed as $\tau/R_t$, $R_t$ can be calculated as follows:

$$R_t = (v'_2 - v_2)/i_{bat}. \tag{4}$$

The time constant $\tau$ was undefined in [10]. However, according to the definition of the resistance–capacitance time constant, complete response of an exponential corresponds to $5\tau$, hence the capacitance can be charged to 95% of the applied voltage. In this study, the $5\tau = t''_2 - t'_2$ is defined. Therefore, substituting the $5\tau = t''_2 - t'_2$ and $C_t = \tau/R_t$ can yield an expression as follows:

$$C_t = (t''_2 - t'_2)/5R_t \tag{5}$$

It is noticeable that the LBP voltage from $v_1$ to $v_2$ has a time delay from $t_2$ to $t'_2$, because the minimum sampling time of the GL240 should be considered. However, this time delay does not affect both $R_t$ and $C_t$ estimations.

### 2.4. Open-Circuit Voltage

$v_{ocv}$ can be regarded as a short circuit in the small-signal analysis. Therefore, the $v_{ocv}$ is irrelevant to the parameter estimation in this study.

### 2.5. Battery Specification and Measurement

In this study, the model number of the LiFePO$_4$ cell is LYS4882160S(3005) (Lyang Energy Tech. Corp., Dongguan, China), and its specifications are listed in Table 3. As shown in Figure 2, the pulse-current peak was set to 17.5 A (0.5 C). For state of charges (SOCs) 30%, 50%, and 70%, the measured voltage–time characteristics are presented in Figure 3. The voltages and times are recorded in Table 4 in line with the definition in Figure 2b. Substitution of Table 4 parameters into (1)–(5) could yield the parameters for the PNGV battery model that listed in Table 5. SOCs 30%, 50%, and 70%, were selected, the reasons explained as follows: First, for the LBP, its equivalent electrical parameters based on the PNGV battery model should be measured to observe the system responses for the low, half, and high SOCs. Second, for the LBPRC, the charge system would result in different frequency responses under the light, medium, and heavy loads.

**Table 3.** Specifications of LiFePO$_4$ battery and LPBRC.

| Description | Specification |
|---|---|
| **Single-Cell** | |
| Model number | LYS4882160S (3005) |
| Charge voltage | 3.65 V |
| Capacity in 1C | 35 Ah |
| **LBP** | |
| Series cell | 8 |
| SOC 30% voltage | 26.92 V (at 0.5C charge) |
| SOC 50% voltage | 27.05 V (at 0.5C charge) |
| SOC 70% voltage | 27.23 V (at 0.5C charge) |
| **LBPRC** | |
| AC input voltage | 230 V$_{rms}$ |
| Line frequency | 60 Hz |
| Maximum output voltage | 30 V |
| Maximum output current | 35 A |
| Maximum output power | 1050 W |

**Table 4.** Records of voltages and times.

| Notation | | SOC 30% | | SOC 50% | | SOC 70% | |
|---|---|---|---|---|---|---|---|
| | | Time (s) | Voltage (V) | Time (s) | Voltage (V) | Time (s) | Voltage (V) |
| $t_1$ | $v_1$ | 0 | 26.23 | 0 | 26.41 | 0 | 26.61 |
| $t_2$ | $v_1$ | 10.00 | 26.23 | 10.00 | 26.23 | 10.00 | 26.23 |
| $t'_2$ | $v_2$ | 10.01 | 26.53 | 10.01 | 26.82 | 10.01 | 27.02 |
| $t''_2$ | $v'_2$ | 13.61 | 26.81 | 11.83 | 26.93 | 11.20 | 27.11 |
| $t_3$ | $v_3$ | 20.31 | 26.92 | 20.28 | 27.05 | 20.35 | 27.23 |
| $t'_3$ | $v_4$ | 20.32 | 26.50 | 20.29 | 26.67 | 20.36 | 26.88 |
| $t_4$ | $v_5$ | 60.00 | 26.25 | 60.00 | 26.43 | 60.00 | 26.63 |

**Table 5.** Parameter value of PNGV battery model.

| SOC | $C_x$ (F) | $R_{oir(st)}$ (mΩ) | $R_{oir(sp)}$ (mΩ) | $R_{oir}$ (mΩ) | $R_t$ (mΩ) | $C_t$ (F) |
|---|---|---|---|---|---|---|
| 30% | 9021.3 | 17.1 | 24.0 | 20.6 | 16.0 | 83.8 |
| 50% | 8995.0 | 23.4 | 21.7 | 22.6 | 6.3 | 58.0 |
| 70% | 9056.3 | 23.4 | 20.0 | 22.2 | 5.1 | 46.7 |
| Average | 9024.3 | 21.3 | 23.9 | 21.8 | 9.1 | 62.8 |

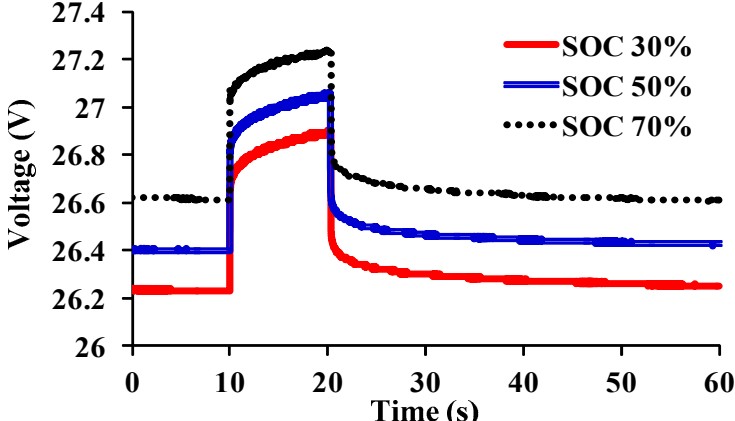

**Figure 3.** Voltage−time characteristics for a LiFePO$_4$ battery pack (LBP) at states of charge (SOCs) 30%, 50%, and 70%.

## 3. Small-Signal System Matrix

Figure 4a depicts the PSFB-PCDR circuit, including the DC input source $V_{inps}$, power switches $Q_a$ to $Q_d$, a blocking capacitance $C_b$, a transformer $T_1$, rectification diodes $D_{f1}$ to $D_{f4}$, current-doubler inductances "$L_{cdr1}$ to $L_{cdr4}$", and an output capacitance $C_o$. The positive and negative electrodes of the LBP are respectively connected to the PSFB-PCDR outlets $o_1$ and $o_2$. Figure 4b presents the operating timing diagram of PSFB-PCDR that includes the driving signals $v_a$ to $v_d$ for $Q_a$ to $Q_d$ and the primary-side voltage $v_{tp}$ across $T_1$; the $T_{sw}$ is the operating switching period for $v_a$ to $v_d$; the $d_y$ represents the PSFB-PCDR operating duty cycle ratio.

In Figure 4, the TTS model can replace the circuit inside the a-frame, as illustrated in Figure 5a. This model has a dependent voltage source $v_r$, two current sources ($i_{r1}$ and $i_{r2}$), and a resistance $R_{eqs}$. The $V_{inps}$ can be reflected to the $T_1$ secondary side becoming $v_r$, as follows:

$$v_r = V_{inps}d_y/n \tag{6}$$

where $n$ is the $T_1$ turns ratio, it equals to the formula $N_p/N_{s1} = N_p/N_{s2}$.

The rising and falling slopes of the $T_1$ input current indicate that they are affected by the PSFB-PCDR operating switching frequency $f_{sw}$ and transformer leakage inductance $L_k$. According to

the method from the [15], using an equivalent resistance $R_{eqs}$ can model the slope change. Therefore, the $R_{eqs}$ is given by

$$R_{eqs} = L_k f_{sw}/(2n^2) \tag{7}$$

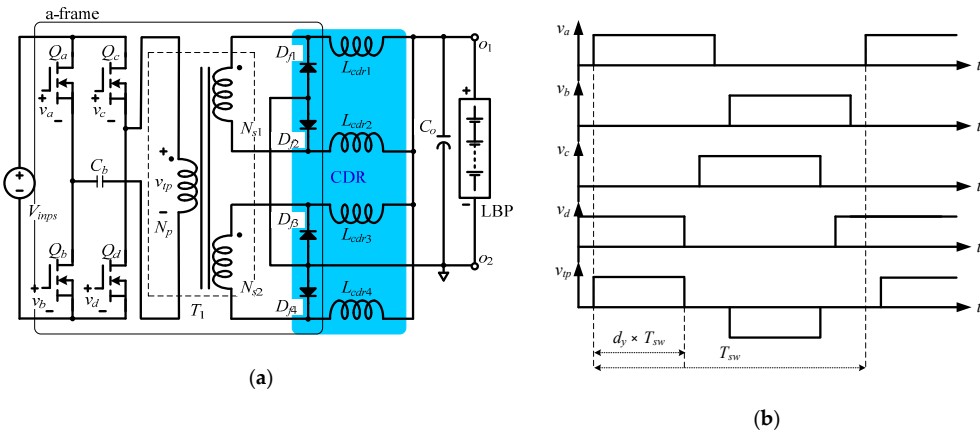

**Figure 4.** PSFB-PCDR circuit schematic and operating timing. (**a**) Circuit scheme. (**b**) Operating timing diagram.

Because of $f_{sw}$, $L_k$ and $n$ are fixative values in the design, and the derivation of the system transfer function will focus on the LPBRC output voltage and current to the operating duty cycle; therefore, using (7) to represent the dynamic influence of $L_k$ can be acceptable.

The PSFB-PCDR secondary-side operations can be regarded as the buck converter, and the $L_{cdr1}$ to $L_{cdr4}$ have the same inductance value; therefore, the four inductances can be treated as parallel connections, and they can be expressed as an inductance $L_{cdr}$. As a result, the output inductance $L_o$ equals to $L_{cdr}/4$ [13,14,26]. Moreover, the $C_o$ contains an equivalent series resistance $R_{esr}$.

A CV or CC power replenishes the LBP by way of the power cable; the wire resistance-inductance would influence the gain and phase of the system frequency response. Therefore, the model considers the wire resistance $R_c$ and inductance $L_c$, which lie between the PSFB-PCDR outlet and LBP. The illustration in Figure 5b is an equivalent circuit.

In Figure 5b, the $R_c$ connects with the $R_{oir}$ in series; hence, $R_c + R_{oir}$ equal to $R_x$. The final equivalent circuit, depicted in Figure 5c, is in accordance with these conditions. Using Kirchhoff's voltage and current laws and the mesh-current method, the loop equations can be obtained as follows:

$$v_r = R_{eqs}i_p + v_{Lo} + R_{esr}(i_p - i_{bat}) + v_{co} \tag{8}$$

$$R_{esr}(i_p - i_{bat}) + v_{co} = v_{Lc} + v_{ct} + R_x i_{bat} + v_{cx} \tag{9}$$

$$i_{Co} = i_p - i_{bat} \tag{10}$$

$$v_{ct} = R_t(i_{bat} - i_{ct}). \tag{11}$$

The LBPRC voltage can be expressed as:

$$v_{bat} = v_{ct} + R_x i_{bat} + v_{cx}. \tag{12}$$

Substitution of (6) and $v_{Lo} = L_o(di_p/dt)$ in (8), $v_{Lc} = L_c(di_{bat}/dt)$ in (9), and $i_{Ct} = C_t(dv_{ct}/dt)$ in (11), the state-space representations can be expressed as follows:

$$di_p/dt = (1/L_o)[(d_y V_{inPS}/n) - (R_{eqs} + R_{esr})i_p + R_{esr}i_{bat} - v_{co}] \tag{13}$$

$$di_{bat}/dt = (1/L_c)[R_{esr}i_p + v_{co} - (R_{esr} + R_x)i_{bat} - v_{ct} - v_{cx}] \tag{14}$$

$$dv_{Co}/dt = (1/C_o)(i_p - i_{bat}) \tag{15}$$

$$dv_{Ct}/dt = (1/C_t)[(i_{bat} - (v_{ct}/R_t)] \tag{16}$$

$$dv_{Cx}/dt = i_{bat}/C_x. \tag{17}$$

The state-space variables can accede to a DC value plus a small-signal perturbation; therefore, substitution of $i_p = I_p + \tilde{i}_p$, $i_{bat} = I_{bat} + \tilde{i}_{bat}$, $v_{Co} = V_{Co} + \tilde{v}_{Co}$, $v_{Ct} = V_{Ct} + \tilde{v}_{Ct}$, and $v_{Cx} = V_{Cx} + \tilde{v}_{Cx}$ into (13)–(17) can yield the small-signal system matrix of PSFB-PCDR as follows:

$$\begin{bmatrix} x_1' \\ x_2' \\ x_3' \\ x_4' \\ x_5' \end{bmatrix} = \mathbf{A} \begin{bmatrix} \tilde{i}_p \\ \tilde{v}_{Co} \\ \tilde{i}_{bat} \\ \tilde{v}_{Ct} \\ \tilde{v}_{Cx} \end{bmatrix} + \mathbf{B}\tilde{d}_y = \begin{bmatrix} a_{11} & a_{12} & a_{13} & a_{14} & a_{15} \\ a_{21} & a_{22} & a_{23} & a_{24} & a_{25} \\ a_{31} & a_{32} & a_{33} & a_{34} & a_{35} \\ a_{41} & a_{42} & a_{43} & a_{44} & a_{45} \\ a_{51} & a_{52} & a_{53} & a_{54} & a_{55} \end{bmatrix} \begin{bmatrix} \tilde{i}_p \\ \tilde{v}_{Co} \\ \tilde{i}_{bat} \\ \tilde{v}_{Ct} \\ \tilde{v}_{Cx} \end{bmatrix} + \begin{bmatrix} b_1 \\ b_2 \\ b_3 \\ b_4 \\ b_5 \end{bmatrix} \tilde{d}_y \tag{18}$$

$$\tilde{v}_{bat} = C \begin{bmatrix} \tilde{i}_p & \tilde{v}_{Co} & \tilde{i}_{bat} & \tilde{v}_{Ct} & \tilde{v}_{Cx} \end{bmatrix}^T = \begin{bmatrix} c_1 & c_2 & c_3 & c_4 & c_5 \end{bmatrix} \begin{bmatrix} \tilde{i}_p & \tilde{v}_{Co} & \tilde{i}_{bat} & \tilde{v}_{Ct} & \tilde{v}_{Cx} \end{bmatrix}^T \tag{19}$$

$$\tilde{i}_{bat} = E \begin{bmatrix} \tilde{i}_p & \tilde{v}_{Co} & \tilde{i}_{bat} & \tilde{v}_{Ct} & \tilde{v}_{Cx} \end{bmatrix}^T = \begin{bmatrix} e_1 & e_2 & e_3 & e_4 & e_5 \end{bmatrix} \begin{bmatrix} \tilde{i}_p & \tilde{v}_{Co} & \tilde{i}_{bat} & \tilde{v}_{Ct} & \tilde{v}_{Cx} \end{bmatrix}^T \tag{20}$$

where $x_1' = d\tilde{i}_p/dt$, $x_2' = d\tilde{v}_{Co}/dt$, $x_3' = d\tilde{i}_{bat}/dt$, $x_4' = d\tilde{v}_{Ct}/dt$, and $x_5' = d\tilde{v}_{Cx}/dt$. The calculations for the elements in matrices **A**, **B**, **C**, and **E** are listed in Table 6.

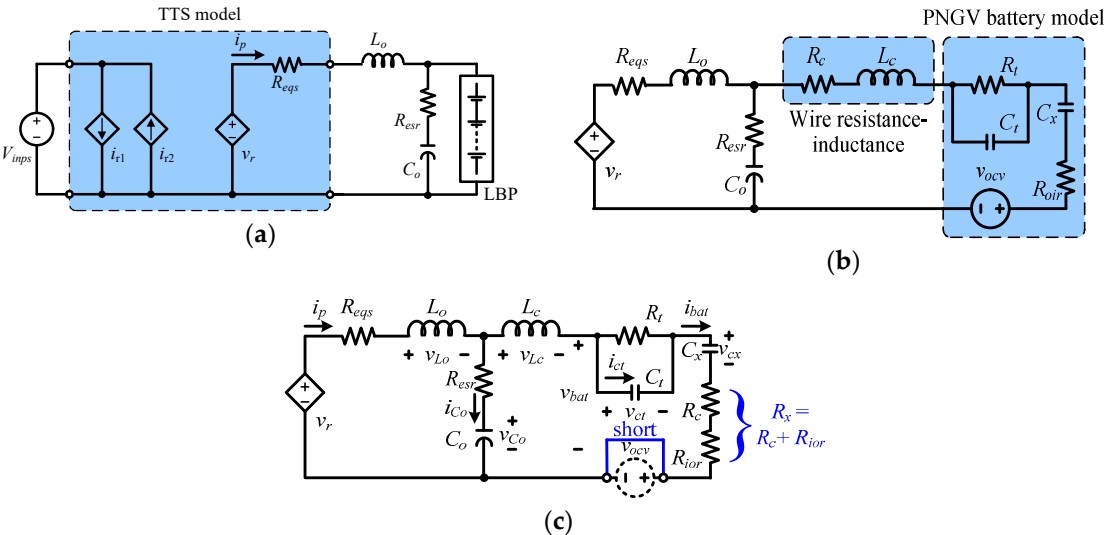

**Figure 5.** PSFB-PCDR equivalent model. (**a**) Three-terminal switch (TTS) model. (**b**) TTS model combines the wire resistance-inductance and the PNGV battery model. (**c**) Final equivalent circuit.

**Table 6.** Element calculations for matrices **A**, **B**, **C**, and **E**.

| Notation | | | | | Equation | | | | |
|---|---|---|---|---|---|---|---|---|---|
| $a_{11}$ | $a_{12}$ | $a_{13}$ | $a_{14}$ | $a_{15}$ | $-(R_{eqs} + R_{esr})/L_o$ | $-1/L_o$ | $R_{esr}/L_o$ | 0 | 0 |
| $a_{21}$ | $a_{22}$ | $a_{23}$ | $a_{24}$ | $a_{25}$ | $1/C_o$ | 0 | $-1/C_o$ | 0 | 0 |
| $a_{31}$ | $a_{32}$ | $a_{33}$ | $a_{34}$ | $a_{35}$ | $R_{esr}/L_c$ | $1/L_c$ | $-(R_{esr} + R_x)/L_c$ | $-1/L_c$ | $-1/L_c$ |
| $a_{41}$ | $a_{42}$ | $a_{43}$ | $a_{44}$ | $a_{45}$ | 0 | 0 | $1/C_t$ | $-1/R_tC_t$ | 0 |
| $a_{51}$ | $a_{52}$ | $a_{53}$ | $a_{54}$ | $a_{55}$ | 0 | 0 | $1/C_x$ | 0 | 0 |
| $b_1$ | $b_2$ | $b_3$ | $b_4$ | $b_5$ | $V_{inps}/nL_o$ | 0 | 0 | 0 | 0 |
| $c_1$ | $c_2$ | $c_3$ | $c_4$ | $c_5$ | 0 | 0 | $R_x$ | 1 | 1 |
| $e_1$ | $e_2$ | $e_3$ | $e_4$ | $e_5$ | 0 | 0 | 1 | 0 | 0 |

## 4. Open-Loop Frequency Response

### 4.1. Open-Loop Gain of Charge Voltage

Figure 6a is the developed charge system's configuration. From Figure 6a, the open-loop block diagram of charge voltage is illustrated in Figure 6b, where the charge voltage gain of PSFB-PCDR is the $G_{psfbv}$ and the PSFB controller gain is the $G_{cltr}$. Using the characteristic equation and matrix, (18) and (19), can yield the following transfer function:

$$G_{psfbv}(s) = \tilde{v}_{bat}/\tilde{d}_y = \mathbf{C}(s\mathbf{I} - \mathbf{A})^{-1}\mathbf{B}. \tag{21}$$

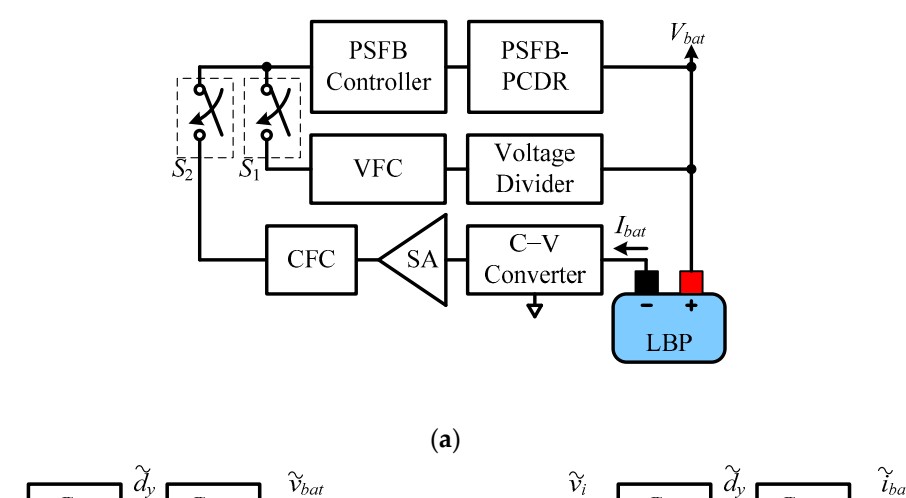

**Figure 6.** Charge system configuration. (**a**) System block diagram. (**b**) Block diagram for open-loop gain of charge voltage. (**c**) Block diagram for open-loop gain of charge current.

Table 7 lists the PSFB-PCDR circuit parameters (Figure 4). Substitution of the $L_k$ = 20 μH, $f_{sw}$ = 100 kHz, $n$ = 14/6 = 2.33 in (7) yields $R_{eqs}$ = 184.2 mΩ and $L_o = L_{cdr1}/4$ = 2.25 μH.

When the DC charge current is 17.5 A, the power cable must sustain this continuous current until LBP replenishment. To avoid the cable temperature rising over 75 °C, in this study, the length and diameter of the power cable were selected as 200 cm and 0.259 cm (No. 10 American wire gauge), respectively; in accordance with the data in [27,28], the $R_c$ = 6.55 mΩ and $L_c$ = 2.91 μH (Figure 5) can be obtained. According to the average parameters in Table 5, the $R_x = R_c + R_{oir}$ = 28.34 mΩ. Substituting these parameters into Table 6 can yield the elements of matrices **A**, **B**, and **C** in (18) and (19).

**Table 7.** Circuit parameters of PSFB-PCDR.

| Notation | Value | Unit |
|----------|-------|------|
| $N_p$ | 14 | turns |
| $N_{s1} = N_{s2} = N_s$ | 6 | turns |
| $L_k$ | 20 | μH |
| $L_{cdr1}-L_{cdr4}$ | 9 | μH |
| $C_o$ | 8200 | μF |
| $C_b$ | 2.2 | μF |
| $R_{esr}$ | 5 | mΩ |
| $V_{inps}$ | 400 | V |
| $f_{sw}$ | 100 | kHz |
| $R_c$ | 6.55 | mΩ |
| $L_c$ | 2.91 | μH |

The characteristics of the operating duty cycle $d_y$ with input voltage $v_i$ is illustrated in Figure 7. Using the MATLAB curve-fitting function, a linear polynomial can be expressed as follows:

$$d_y = 0.15v_i + 0.025 \tag{22}$$

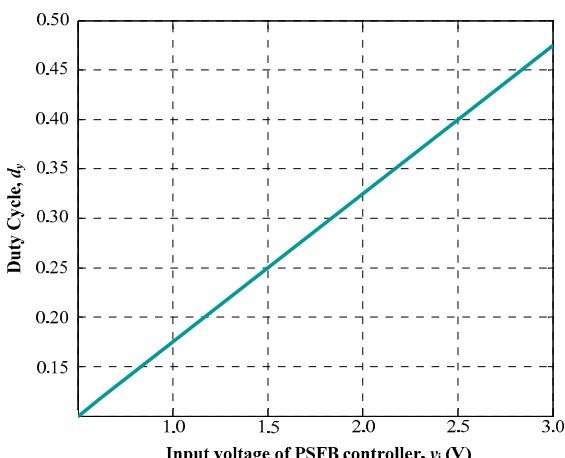

**Figure 7.** Relationship of operating duty cycle $d_y$ to input voltage $v_i$.

Substitution of $v_i = V_i + \widetilde{v}_i$ and $d = D_y + \widetilde{d}_y$ in (22) yields the PSFB controller gain as follows:

$$G_{cltr}(s) = \widetilde{d}_y/\widetilde{v}_i = 0.15 \tag{23}$$

According to (21) and (23), the transfer function of the charge voltage in open-loop condition can be obtained as follows:

$$G_{ov}(s) = \widetilde{v}_{bat}/\widetilde{v}_i = G_{cltr}(s)G_{psfbv}(s) \tag{24}$$

Using MATLAB, the frequency response simulation of $G_{ov}$ (s) was completed and plotted in Figure 8. In this Bode plot, the low-frequency gain (LFG) was 11.4 dB, and the bandwidth (BW) was 237 Hz. The frequency and phase at 0 dB were 1.7 kHz and $-49.9°$, respectively. Moreover, in this study, the TTS model with the PNGV battery model in the LBPRC application is proposed for the first time, along with the complete design procedures for the CV and CC feedback compensations. Therefore, the gain and phase slopes in three bands 0.1 to 1 kHz, 1 to 10 kHz, and 10 to 100 kHz, can be observed to compare and confirm the proposed model, which is feasible. At 0.1 to 1 kHz, the gain and phase slopes were $-10.85$ dB/decade and $-31.4°$/decade, respectively. At 1 to 10 kHz, the gain and phase slopes were 6.15 dB/decade and $15.7°$/decade; at 10 to 100 kHz, the gain and phase slopes were $-27.3$ dB/decade and $-92.5°$/decade.

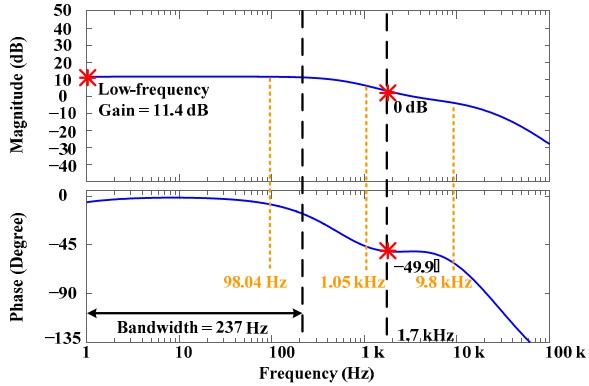

**Figure 8.** Bode plot simulation: $G_{ov}$ (s).

Furthermore, break frequencies included $f_{opv1}$ = 98.04 Hz (pole), $f_{ozv1}$ = 1.05 kHz (zero), and $f_{opv2}$ = 9.8 kHz (pole). From Figure 8, several circumstances could be observed:

(1) The steady-state error of charge voltage would be severe during the feedback operation, because the low-frequency gain of 11.4 dB was too low.
(2) A narrow bandwidth of 237 Hz was observed.
(3) Two poles ($f_{opv1}$ and $f_{opv2}$) and one zero ($f_{ozv1}$) can be found in the Bode plot.

*4.2. Comparison of Frequency Response Using the Different SOC Parameters and the Battery Model*

From Table 5, different SOC parameters resulted in different frequency responses of $G_{ov}$ (s), as illustrated in Figure 9a. Expanding the magnitude and phase scales from Figure 9a, it could be observed that three SOC (30%, 50%, and 70%) frequency responses approximated to the average result (blue solid-line). Therefore, the proposed PNGV battery model in this study can employ the average value (Table 5) to analyze and simulate the system Bode plot.

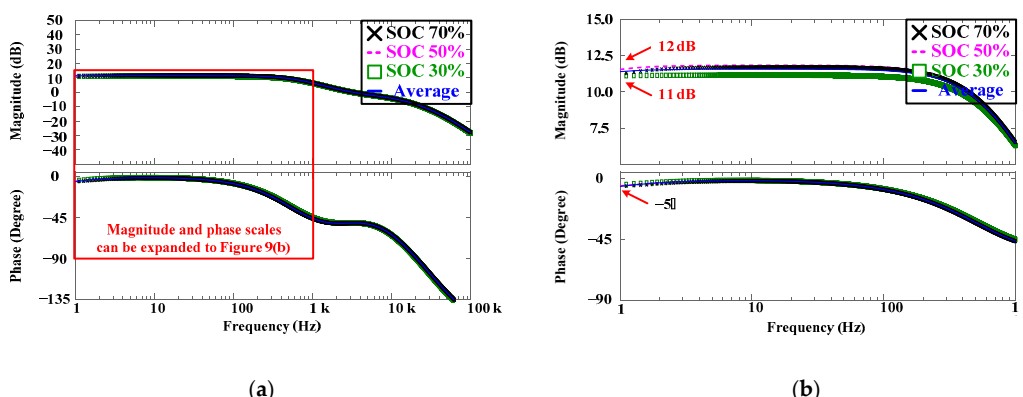

(**a**)                                                                 (**b**)

**Figure 9.** Different SOCs of $G_{ov}$ (s). (**a**) Using the three SOCs and the average parameters. (**b**) Expanding scale from Figure 9a.

To compare the frequency response discrepancies in the PNGV model, the RC model, and the no-wire resistance-inductance, these simulations presented in Figure 10. From Figure 10, several circumstances can be observed that are described in the following.

(1) No-wire resistance-inductance: The $R_c$ and $L_c$ (Figure 5) were neglected; therefore, the $G_{ov}$ (s) from (24) could be replaced with the $G_{ov1}$ (s). The phase of $G_{ov1}$ (s) deviated the $G_{ov}$ (s) seriously, and the gain of $G_{ov1}$ (s) was less than the $G_{ov}$ (s), the band ranges were from 20 to 100 kHz (Figure 10).

(2) Using the RC model: The $R_{esr}$, $R_c$, $L_c$, $R_t$, and $C_t$ (Figure 5) were neglected; therefore, the $G_{ov}$ (s) from (24) could be replaced with $G_{ov2}$ (s). The gain and phase slopes for the $G_{ov2}$ (s) could not present the break frequency changes, such as the pole-zero of $G_{ov}$ (s).

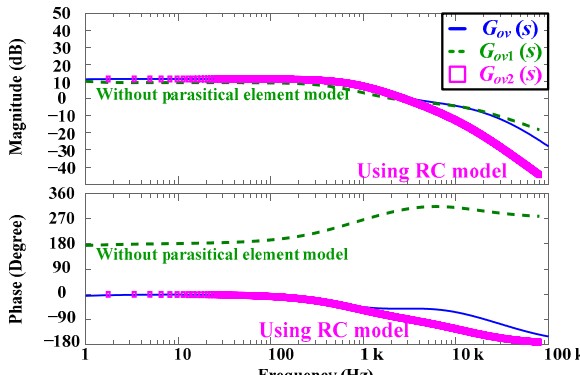

**Figure 10.** Frequency response simulation using different battery models.

From these simulations, the stability analysis and the pole-zero compensation were difficult to implement for the LBPRC design. Therefore, the LBPRC analysis adopting the PNGV battery model is a feasible method to obtain the system frequency response, and further to design the VFC and CFC for the CV and CC charges. The Bode plot of $G_{ov}$ (s) in practice will be measured; both simulation and measurement correspond to the anticipated result of the theory.

*4.3. Open-Loop Gain of Charge Current*

From Figure 6a, the open-loop block diagram of charge current is illustrated in Figure 6c, the $G_{psfbc}$ represents the charge current gain of PSFB-PCDR. Using the characteristic equation and matrix, (18) and (20), can obtain the following transfer function:

$$G_{psfbc}(s) = \widetilde{i}_{bat}/\widetilde{d}_y = \mathbf{E}(s\mathbf{I} - \mathbf{A})^{-1}\mathbf{B} \tag{25}$$

where the elements of matrix **E** can be obtained from Table 6. According to (23) and (25), the transfer function of charge current in open-loop condition can be obtained as follows:

$$G_{oc}(s) = \widetilde{i}_{bat}/\widetilde{v}_i = G_{cltr}(s)G_{psfbc}(s) \tag{26}$$

Using MATLAB, the frequency response simulation of $G_{oc}$ (s) was completed and plotted in Figure 11. In this Bode plot, the low-frequency gain was 35.6 dB, and the bandwidth was 200 Hz. The frequency and phase at 0 dB were 50 kHz and $-140°$, respectively. Moreover, at 0.1 to 1 kHz, the gain and phase slopes were $-10.8$ dB/decade and $-30.2°$/decade; at 1 to 10 kHz, the gain and phase slopes were 7.7 dB/decade and 15.5°/decade, respectively; at 10 to 100 kHz, the gain and phase slopes were $-4.9$ dB/decade and $-92.3°$/decade, respectively. Furthermore, break frequencies included the $f_{opc1} = 98.04$ Hz (pole), $f_{ozc1} = 1.05$ kHz (zero), and $f_{opc2} = 9.8$ kHz (pole). From Figure 11, three circumstances can be observed that are described in the following.

(1) A narrow bandwidth of 200 Hz was observed.
(2) The phase margin was less than $-135°$ (at 50 kHz), which has acceded to the allowable tolerance of $45°$ [ $|-180°-(-135°)|$ ]; therefore, the charge current loop was regarded as unstable.
(3) Two poles ($f_{opc1}$ and $f_{opc2}$) and one zero ($f_{ozc1}$) can be found in the Bode plot.

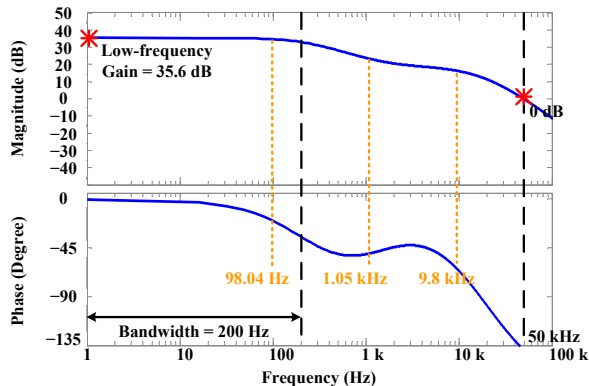

**Figure 11.** Bode plot simulation: $G_{oc}$ (s).

## 5. Open-Loop Frequency Response

Design considerations and procedures will be discussed in the following.

### 5.1. VFC for CV Charge

According to the simulation of $G_{ov}$ (Figure 8), the VFC design is as follows:

(1)　The low-frequency gain must be heightened to reduce the steady-state error of charge voltage.
(2)　The system bandwidth should be extended to accelerate the response speed.
(3)　The phase margin should be greater than $-135°$ to ensure the stability of the CV charge loop.

In Figure 6a, the voltage divider and VFC circuit are illustrated as Figure 12a, including voltage divider resistances ($R_{vd1}$ and $R_{vd2}$), an operational amplifier OP$_1$, a charge voltage reference command $V_{refv}$, resistances ($R_{fv1}$ and $R_{fv2}$), and capacitances ($C_{fv1}$ and $C_{fv2}$). The control block diagram of charge voltage in closed-loop operation is depicted in Figure 12b. The $k_v$ is a constant that can be calculated by the formula $R_{vd2}/(R_{vd1}+R_{vd2})$. The VFC gain is $G_{vfc}$, whose transfer function is given by

$$G_{vfc}(s) = \tilde{v}_{vfco}/\tilde{v}_{vfci} = (1 + sR_{fv2}C_{fv1})/\left\{ [sR_{fv1}(C_{fv1}+C_{fv2})][1 + (sR_{fv2}C_{fv1}C_{fv2})/(C_{fv1}+C_{fv2})] \right\} \quad (27)$$

and its corner frequencies at the pole and zero, both are given in

$$f_{zv} = 1/(2\pi R_{fv2}C_{fv1}) \quad (28)$$

$$f_{pv} = 1/(2\pi R_{fv2}C_{fv2}). \quad (29)$$

Moreover, $g_{av}$ represents the gain, it can be expressed as:

$$g_{av} = R_{fv2}/R_{fv1}. \quad (30)$$

From Figure 12b, a transfer function of the open-loop voltage is

$$G_v(s) = k_v G_{ov}(s). \quad (31)$$

Substitution of $k_v$ = 0.073 and (24) in (31) yields $G_v$ (s), whose frequency response simulation is plotted in Figure 13. According to this Bode plot, the VFC design procedures can be performed to compensate the charge voltage loop. They are described in the following.

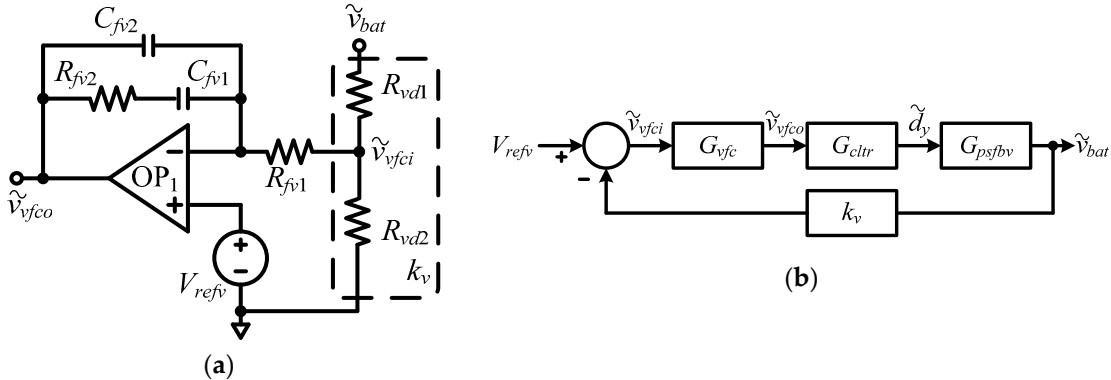

**Figure 12.** Charge voltage loop. (**a**) Voltage divider and voltage feedback controller (VFC) circuit. (**b**) Control block diagram of closed-loop operation.

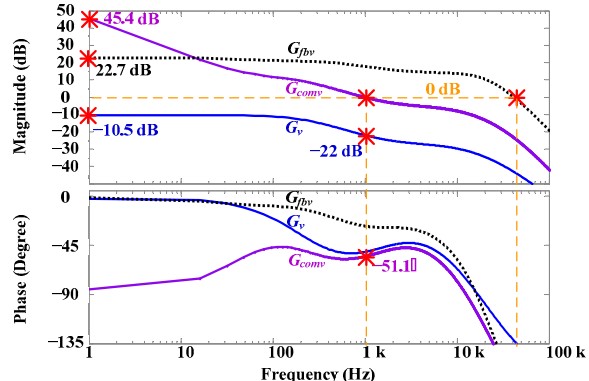

**Figure 13.** Bode plot simulations: $G_v$ (*s*), $G_{comv}$ (*s*), and $G_{fbv}$ (*s*).

### 5.1.1. VFC Design Procedure

Step 1: Setting crossover frequency

In the switching power supply applications, the crossover frequency range can be determined between $f_{sw}/100$ to $f_{sw}/10$. In this study, we considered the PNGV battery model for obtaining the frequency responses of $G_{ov}$ (*s*) and $G_{oc}$ (*s*). Therefore, the aforementioned crossover frequency range can be directly applied in the LBPRC design.

The crossover frequency of the charge voltage loop, $f_{cov}$, can be set at the minimum frequency 1 kHz ($f_{cov} = f_{sw}/100 = 100k/100$), because the voltage–time change rate of charge voltage is slow. Moreover, to prevent the resonant factor from influencing the system stability, the $f_{cov}$ should be kept away from the output filter resonant frequency ($f_{opv1} = 98.01$ Hz); therefore, the frequency ranges can be expressed as $f_{opv1} << f_{cov}$, as depicted in Figure 14.

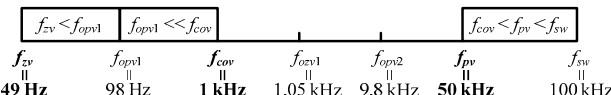

**Figure 14.** Frequency point setting of charge voltage loop.

Step 2: Gain heave at the crossover frequency

According to Figure 13, at 1 kHz, the $G_v$ should be hove from $-22$ to 0 dB; hence, the $g_v$ and $R_{fv1}$ of (30) can be set to 12.6 and 10 kΩ, respectively, and then $R_{fv2} = 126$ kΩ can be obtained.

Step 3: Pole break frequency of VFC

To reduce the charge voltage ripple, the pole break frequency $f_{pv}$ must be less than $f_{sw}$. Moreover, the phase margin should be increased if the $f_{pv}$ was greater than $f_{cov}$. Therefore, substitution of the $f_{pv} = f_{sw}/2 = 50$ kHz and $R_{fv2} = 126$ kΩ into (29) can yield $C_{fv2} = 25.26$ pF. Under these conditions, the frequency range can be expressed as $f_{cov} < f_{pv} < f_{sw}$, as depicted in Figure 14.

Step 4: Zero break frequency of VFC

To prevent the phase shift from being less than $-180°$, the zero break frequency $f_{zv}$ should be less than $f_{opv1}$. Thus, substitution of the $f_{zv} = f_{opv1}/2 = 49$ Hz and the $R_{fv2} = 126$ kΩ into (28) can yield $C_{fv1} = 25.78$ nF. Under these conditions, the frequency range can be expressed as $f_{zv} < f_{opv1}$, as depicted in Figure 14.

Substitution of the $R_{fv1} = 10$ kΩ, $R_{fv2} = 126$ kΩ, $C_{fv1} = 25.78$ nF, and $C_{fv2} = 25.26$ pF into (27) and (31) could yield the transfer function of open-loop voltage gain for the LBPRC with the VFC, $G_{comv}$ (s), the frequency response simulation was plotted in Figure 13. From Figure 13, the low-frequency gain of $G_{comv}$ (s) was increased to 45.4 dB; at crossover frequency $f_{cov} = 1$ kHz, the phase was $-51.1°$, which was greater than $-135°$ and was over the tolerance of $45°$ ($|-180° - 135°| = 45°$), hence the proposed VFC design could meet the stability requirement.

Moreover, the transfer function of closed-loop voltage gain, $G_{fbv}$ (s), was simulated in Figure 13. From Figure 13, the feedback operation for the charge voltage loop could observe that the low-frequency gain of $G_{fbv}$ (s) was 22.7 dB; the phase was greater than $-135°$ when the band ranges were from 1 to 26.8 kHz.

From Figure 13, the simulation could comprehend that the LBPRC incorporating the VFC could address the stable requirement for the CV charge. Two reasons explain as follows:

(1) Determining the system stability from the phase of $G_{comv}$ (s) is a critical method which has been mentioned in previous studies [29,30] for the design of switching power supplies. Because the phase $-51.1°$ of $G_{comv}$ (s) at 0 dB was greater than $-135°$, a stable charge voltage loop could be fulfilled.

(2) It was a rational condition that the phase of $G_{fbv}$ (s) was less than $-135°$ at 0 dB, because the negative feedback operation may become unstable influencing the system stability in the high frequencies [31–33]. However, from Figure 13, the bandwidth of $G_{fbv}$ (s) was extended; this result was one of the advantages when the LBPRC operated in the negative feedback mode [32,33].

*5.2. CFC for CC Charge*

According to the simulation of $G_{oc}$ (Figure 11), the CFC design consideration is described as follows:

(1) The low-frequency gain must be heightened to reduce the steady-state errors of charge current.
(2) The 0 dB frequency should be decreased because the initial 50 kHz was higher than the suggested frequency of 10 kHz ($f_{sw}/10$).
(3) The phase margin should be greater than $-135°$ to ensure the stable operation for the CC charge loop.

From Figure 6a, the signal amplifier (SA) and CFC circuit are illustrated in Figure 15a, including the current shunt $R_s$, an operational amplifier OP$_2$, a charge current reference command $V_{refc}$, resistances ($R_{fc1}$ and $R_{fc2}$), and capacitances ($C_{fc1}$ and $C_{fc2}$), hence using the OP$_2$, $V_{refc}$, $R_{fc1}$, $R_{fc2}$, $C_{fc1}$, and $C_{fc2}$, can compose an error amplifier to implement a proportional-integral control. The control block diagram of charge current in the closed-loop operation is depicted in Figure 15b. The $k_c$ is a constant that can be calculated by the $R_s \times g_{sa}$ (DC gain of SA). The CFC gain is $G_{cfc}$, whose transfer function is given by

$$G_{cfc}(s) = \widetilde{v}_{cfco}/\widetilde{v}_{cfci} = (1 + sR_{fc2}C_{fc1})/\left\{[sR_{fc1}(C_{fc1} + C_{fc2})][1 + (sR_{fc2}C_{fc1}C_{fc2})/(C_{fc1} + C_{fc2})]\right\}, \quad (32)$$

and its corner frequencies at the pole and zero are both given in

$$f_{zc} = 1/(2\pi R_{fc2}C_{fc1})$$  (33)

$$f_{pc} = 1/(2\pi R_{fc2}C_{fc2}).$$  (34)

Moreover, $g_c$ represents the gain, it can be expressed as:

$$g_c = R_{fc2}/R_{fc1}$$  (35)

From Figure 15b, the open-loop transfer function is

$$G_c(s) = k_c G_{oc}(s)$$  (36)

Substitution of the $k_c$ = 0.097 and (26) in (36) can yield $G_c$ (s), whose frequency response simulation is plotted in Figure 16. According to this Bode plot, the CFC design procedure can be performed to compensate charge current loop. They are described in the following.

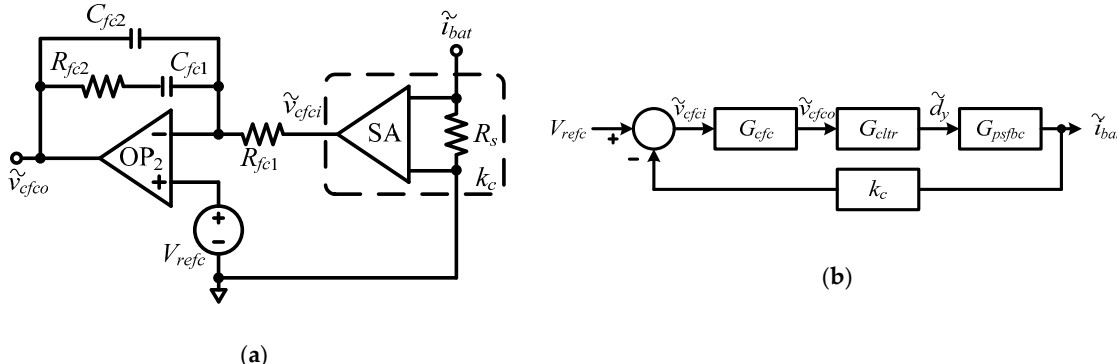

(**a**)                                    (**b**)

**Figure 15.** Charge current loop. (**a**) SA and current feedback controller (CFC) circuit. (**b**) Control block diagram of closed-loop operation.

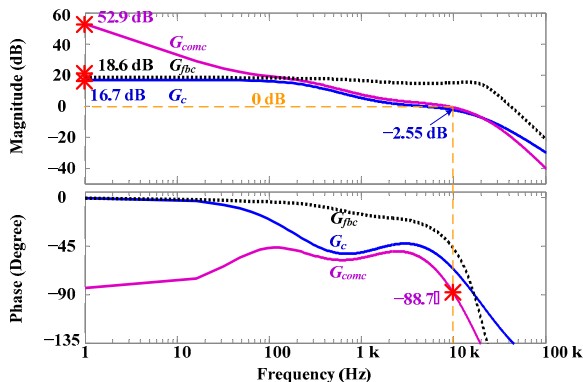

**Figure 16.** Bode plot simulations: $G_c$ (s), $G_{comc}$ (s), and $G_{fbc}$ (s).

### 5.2.1. CFC Design Procedure

**Step 1: Setting crossover frequency**

The crossover frequency of charge current loop, $f_{coc}$, can be set to the minimum frequency 10 kHz ($f_{coc} = f_{sw}/10 = 100k/10$) because the current–time change rate of charge current is fast. Moreover, to prevent the resonant factor from influencing the system stability, the $f_{coc}$ should be kept away from the output filter resonant-frequency ($f_{opc1}$ = 98.01 Hz); therefore, the frequency ranges can be expressed as $f_{opc1} \ll f_{coc}$, as depicted in Figure 17a.

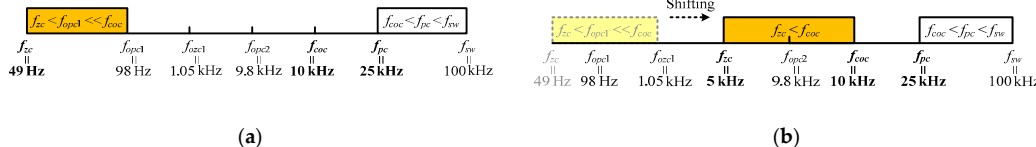

**Figure 17.** Frequency point design of charge current loop. (**a**) Normal PI design. (**b**) Using the zero break frequency shift (ZBFS) method.

Step 2: Gain heave at the crossover frequency

According to Figure 16, at 10 kHz, the $G_c$ should to be heaved from -2.25 to 0 dB. Therefore, the $g_c$ and $R_{fc1}$ in (35) can be set to 1.296 and 10 kΩ, respectively. The $R_{fc2}$ = 12.96 kΩ can then be obtained.

Step 3: Pole break frequency of CFC

To reduce charge current ripple, the pole break frequency $f_{pc}$ must be less than $f_{sw}$. Moreover, the phase margin should be increased if the $f_{pc}$ is greater than $f_{coc}$. Therefore, substitution of the $f_{pc} = f_{sw}/4$ = 25 kHz and $R_{fv2}$ = 12.96 kΩ into (34) can yield $C_{fc2}$ = 491.22 pF. Under these conditions, the frequency range can be expressed as $f_{coc} < f_{pc} < f_{sw}$, as depicted in Figure 17a.

Step 4: Zero break frequency setting of CFC

To prevent the phase shift from being less than $-180°$, the zero break frequency, $f_{zc}$, should be less than $f_{opc1}$. Thus, substitution of the $f_{zc} = f_{opc1}/2$ = 49 Hz and $R_{fc2}$ = 12.96 kΩ into (33) can yield $C_{fc1}$ = 250.62 nF. Under these conditions, the frequency range can be expressed as $f_{zc} < f_{opc1}$, as depicted in Figure 17a.

Substitution of the $R_{fc1}$ =10 kΩ, $R_{fc2}$ = 12.96 kΩ, $C_{fc1}$ = 250.62 nF, and $C_{fc2}$ = 491.22 pF into (32) and (36) could yield the transfer function of open-loop gain for the LBPRC with the CFC, $G_{comc}$ ($s$), whose frequency response simulation was plotted in Figure 16. From Figure 16, the low-frequency gain was increased to 52.9 dB. At the $f_{coc}$ = 10 kHz, the phase was $-88.7°$, which was greater than $-135°$. Moreover, the transfer function of the negative feedback operation for the charge current loop was $G_{fbc}$ ($s$), whose low-frequency gain was 18.6 dB; the phase was greater than $-135°$ when the band ranges were from 1 to 26.8 kHz; therefore, the CC charge could address the stability requirement.

### 5.2.2. Current Overshoot Mitigation Using Zero Break Frequency Shift (ZBFS)

As shown in Figure 16, the $G_{fbc}$ gain presented a lapse shape from 1 to 20 kHz. This situation would influence the response speeds of charge current loop, resulting in the current overshoot $i_{oc}$ of the start-up phase and a long settling time, as illustrated in Figure 18a. According to the step 4 of Section 5.2.1., because the $f_{zc}$ = 49 Hz, a large capacitance $C_{fc1}$ = 250.62 nF was obtained; in consequence, the long-time integral action of PI control became a problem to limiting the CFC response speed. However, shifting $f_{zc}$ to a high frequency was an effective method that could resolve this problem, as depicted in Figure 18b; therefore, the $f_{zc}$ was increased to 5 kHz from the zero break frequency of 49 Hz, and then the frequency range could be expressed as $f_{zc} < f_{coc}$, as depicted in Figure 17b. Substitution of the $f_{zc}$ = 5 kHz and $R_{fc2}$ = 12.96 kΩ into (33) yielded $C_{fc1}$ = 2.46 nF.

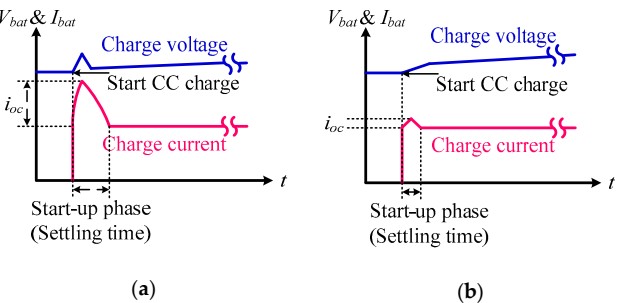

**Figure 18.** During start-up phase, different control technologies result in current overshoot occurrence. (**a**) Normal PI design. (**b**) Using the ZBFS method.

### 5.2.3. Current Overshoot Mitigation Using PSPI Control

Figure 19a presents a PSPI configuration and a practical circuit schema that can be applied to mitigate the charge current overshoot during the LBPRC start-up phase. The $R_s$ and SA are used to detect and amplify the charge current signal $v_{ibat}$, then the SA generates a voltage signal $v_{cfci}$; thus, the CFC compares $v_{cfci}$ with a charge current reference command $I_{o(ref)}$ to generate an error voltage $v_{cfco}$ undergoing proportional (P) or PI compensation to control the PSFB controller (Figure 6).

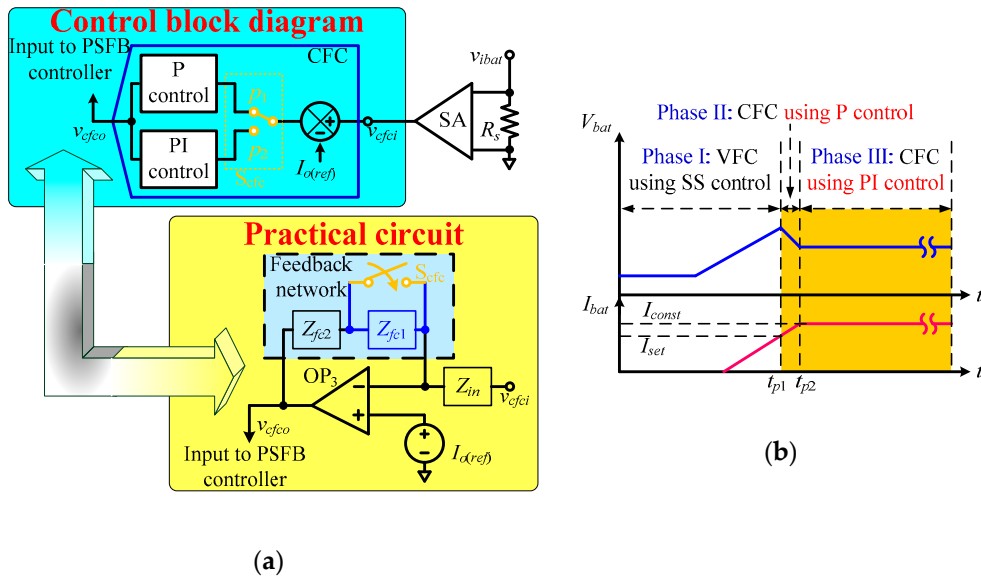

(**a**)

(**b**)

**Figure 19.** A novel PSPI control. (**a**) PSPI configuration diagram and practical circuit. (**b**) Phases I to III.

From Figure 19b, during Phase I, the LBPRC is dominated by the VFC with the soft-start (SS) control, and the $V_{bat}$ slowly increases, thus avoiding the voltage overshoot occurrence at the LBPRC output side. As the $I_{bat}$ increases to reach the current setting for the $I_{set}$, the shifting component $S_{cfc}$ (Figure 19a) is connected to the position $p_1$, enabling the CFC to implement P control during the Phase II; hence, the CFC response speed can be enhanced to prevent the charge current overshoot. In the Phase III, the $I_{bat}$ reaches the target charge CC $I_{const}$, the $S_{cfc}$ is sifted to the position $p_2$, and then the CFC implements the PI control to stabilize the LBP charge current for achieving the CC charge.

Figure 19a shows the CFC circuit with PSPI control composed of an operational amplifier OP$_3$, compensative elements ($Z_{in}$, $Z_{fc1}$, and $Z_{fc2}$), and the $I_{o(ref)}$. Using $Z_{in}$ and $Z_{fc2}$ can yield the P gain to implement the P control; the $Z_{in}$ combines $Z_{fc1}$ with $Z_{fc2}$ can achieve the PI control.

Moreover, the shifting certification for the $S_{cfc}$ is defined as follows:

$$S_{cfc} = \begin{cases} \text{cut-in, when } v_{cfco} \geq v_{cfci} \\ \text{cut-off, when } v_{cfco} < v_{cfci} \end{cases}$$

During Phases I and II, the $v_{cfco}$ is greater than the $V_{cfci}$ to cut in the $S_{cfc}$, the integral component $Z_{fc1}$ is invalid and the unique P control can be implemented through $Z_{fc2}$ and $Z_{in}$ for the CFC. In Phase III, the $S_{cfc}$ is cut off, thus the CFC can employ the $Z_{in}$, $Z_{fc1}$, and $Z_{fc2}$ to implement the PI control. Therefore, from the time interval $[t_{p1}:t_{p2}]$ (Figure 19), the CFC control can shift from P to PI; this control method is effective to mitigate the charge current overshoot. Moreover, the designing parameters from Sections 5.2.1 and 5.2.2 can completely apply to $Z_{in}$, $Z_{fc1}$, and $Z_{fc2}$ without readjusting and redesigning.

### 5.3. LBPRC Charge Strategy

In practice, the SOC and state of health (SOH) of the battery should be evaluated by the battery management system (BMS). This is because the LBP electrical-chemical characteristics would be

affected under different operating conditions. According to the SOC and SOH, the BMS can manipulate the LBPRC to implement a CV or CC charge. Therefore, an optimum charge strategy can be determined by the BMS, rather than the LBPRC. Supplying a stable CV or CC power to charge the LBP is the principal responsibility of the LBPRC.

Because of the LBPRC should be designed as an ideal voltage or current source to replenish LBPs; therefore, this study focused on the VFC design for the CV output operation of LBPRC, and the CFC design for the CC output operation of LBPRC. In [16–19], the CC–CV charge strategy was a suitable method for the LiFePO$_4$ battery charge. Therefore, the CC–CV charge strategy is the first case of charge strategy. Moreover, to fulfill the battery charge requirement, the LBP can be rapidly charged by the LBPRC within one hour. The second case of charge strategy can use the CC to replenish the LBP. Charge conditions of both cases are defined as follows:

Case 1: When the SOC of the LBP approximates to zero, the voltage of the empty LBP is approximately 22 V; if the LBP SOC ranges from empty to 20% (LBP voltage is 27.2 V), the LBP is charged by a 35-A (1C) CC. If the LBP SOC is higher than 20%, then the LBPRC can operate in the CV output mode to charge the LPB. This case can confirm that LBPRC supplying the CV power to charge the LBP is feasible for low to high LPB SOCs. This charge profile will be presented.

Case 2: The LBP SOC ranges from empty to 83% (LBP voltage is 29.2 V). The LBP is charged by a 35-A CC. This case can confirm that LBPRC supplying the CC power to charge the LBP is feasible for low to high LPB SOCs. This charge profile will be presented.

## 6. Experimental Results

The measurement system of frequency response for the LBPRC is depicted in Figure 20. This system comprised a computer, a USB/GPIB interface (National Instruments Inc., Austin, TX, USA), and a frequency response analyzer (FRA) MODEL3120 (Veneable Corp., Austin, TX, USA). The FRA setting parameter and data can be controlled and transmitted through the USB/GPIB interface. Therefore, the fluctuant small signals $\tilde{v}_i$, $\tilde{v}_{vfci}$, and $\tilde{v}_{cfci}$ generating by the FRA can be respectively output to the PSFB controller and adders ($A_1$ and $A_2$). Subsequently, the fluctuant charge voltage $\tilde{v}_{bat}$ and current $\tilde{i}_{bat}$ can be recorded and transmitted to the FRA. Finally, the Bode plot is presented on the computer monitor.

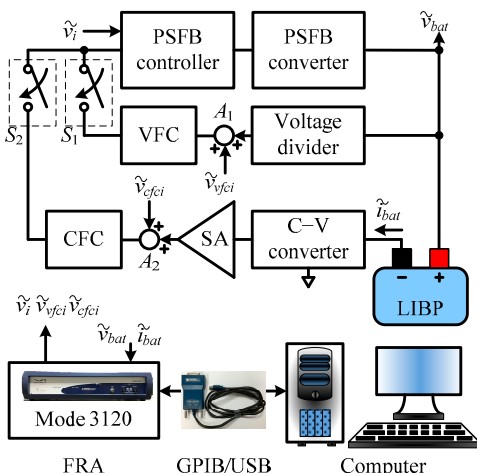

**Figure 20.** Measurement system of frequency response for LBPRC.

Frequency response measurement of open-loop charge voltage: As indicated in Figure 20, when both $S_1$ and $S_2$ were turned off, the FRA output was $\tilde{v}_i$; the fluctuant $\tilde{v}_{bat}$ could then be measured and input to the FRA. The Bode plot of $G_{ov}$ was displayed in Figure 21a. The low-frequency gain was 10 dB (simulation: 11.4 dB), and the bandwidth was 250 Hz (simulation: 237 Hz). At gain of 0 dB, the frequency and phase were 1.6 kHz (simulation: 1.7 kHz) and $-58°$ (simulation: $-49.9°$),

respectively. Moreover, for the break frequencies, the $f_{opc1}$ = 100 Hz had a pole (simulation: 98.04 Hz), the $f_{ozc1}$ = 1.5 kHz had a zero (simulation: 1.05 Hz), and the $f_{opc2}$ = 12 kHz had a pole (simulation: 9.8 kHz). The gain and phase slopes in the different bands were listed in Table 8.

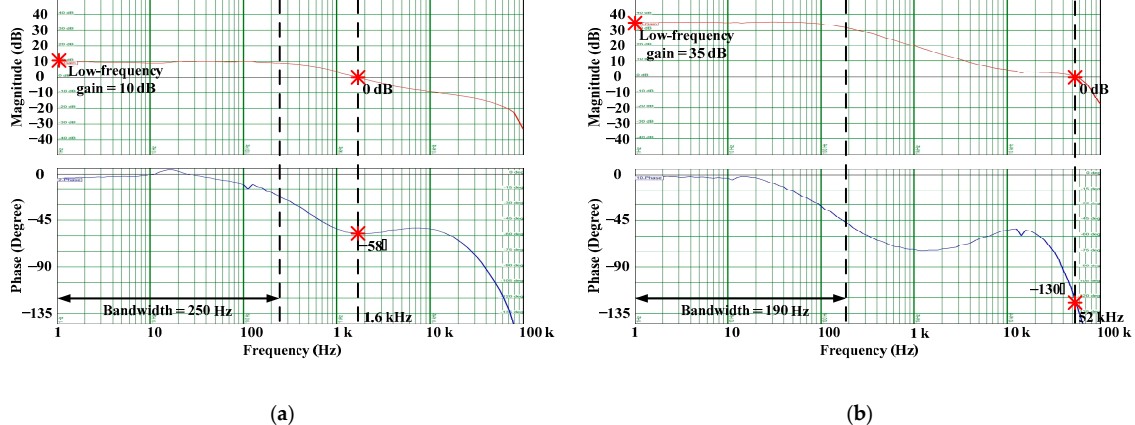

(**a**)　　　　　　　　　　　　　　　　　　　　　　　　　(**b**)

**Figure 21.** Bode plot measurements: (**a**) $G_{ov}$ and (**b**) $G_{oc}$.

**Table 8.** Gain and phase slopes.

|  | 0.1 to 1 kHz | | 1 to 10 kHz | | 10 to 100 kHz | | |
|---|---|---|---|---|---|---|---|
|  | Measurement | Simulation | Measurement | Simulation | Measurement | Simulation | Unit |
| **Open-Loop Charge Voltage** | | | | | | | |
| Gain slope | −7 | −10.85 | −7 | −6.15 | −25 | −27.3 | dB/decade |
| Phase slope | −37 | −31.4 | −10 | −15.7 | −100 | −92.5 | degree/decade |
| **Open-Loop Charge Current** | | | | | | | |
| Gain slope | −15 | −11.4 | −13 | −9 | −10 | −6 | dB/decade |
| Phase slope | −40 | −30.2 | 18 | 15.5 | −95 | −92.3 | degree/decade |

Frequency response measurement of open-loop charge current: As indicated in Figure 20, when both $S_1$ and $S_2$ were turned off, the FRA output was $\widetilde{v}_i$; the fluctuant $\widetilde{i}_{bat}$ could then be measured and input to FRA. The Bode plot of $G_{oc}$ was displayed in Figure 21b. The low-frequency gain was 35 dB (simulation: 35.6 dB), and the bandwidth was 190 Hz (simulation: 200 Hz). At gain of 0 dB, the frequency and phase were 52 kHz (simulation: 50 kHz) and −130° (simulation: −140°), respectively. Moreover, for the break frequencies, the $f_{opv1}$ = 100 Hz had a pole (simulation: 98.04 Hz), the $f_{ozv1}$ = 3.5 kHz had a zero (simulation: 1.05 kHz), and $f_{opv2}$ = 15 kHz had a pole (simulation: 9.8 kHz). The gain and phase slopes in the different bands were listed in Table 8. These measurements demonstrate that the proposed equivalent models are suitable for establishing the small-signal transfer functions of LBPRC.

Frequency response measurement of closed-loop charge voltage: As shown in Figure 20, with the $S_1$ turned on and the $S_2$ turned off, the FRA generated $\widetilde{v}_{vfci}$, which was input to $A_1$, the $\widetilde{v}_{bat}$ could then be measured by the FRA. The Bode plot of $G_{fbv}$ is displayed in Figure 22a. The low-frequency gain was 22 dB (simulation: 22.7 dB). At the gain 0 dB, its frequency was 30 kHz (simulation: 40 kHz).

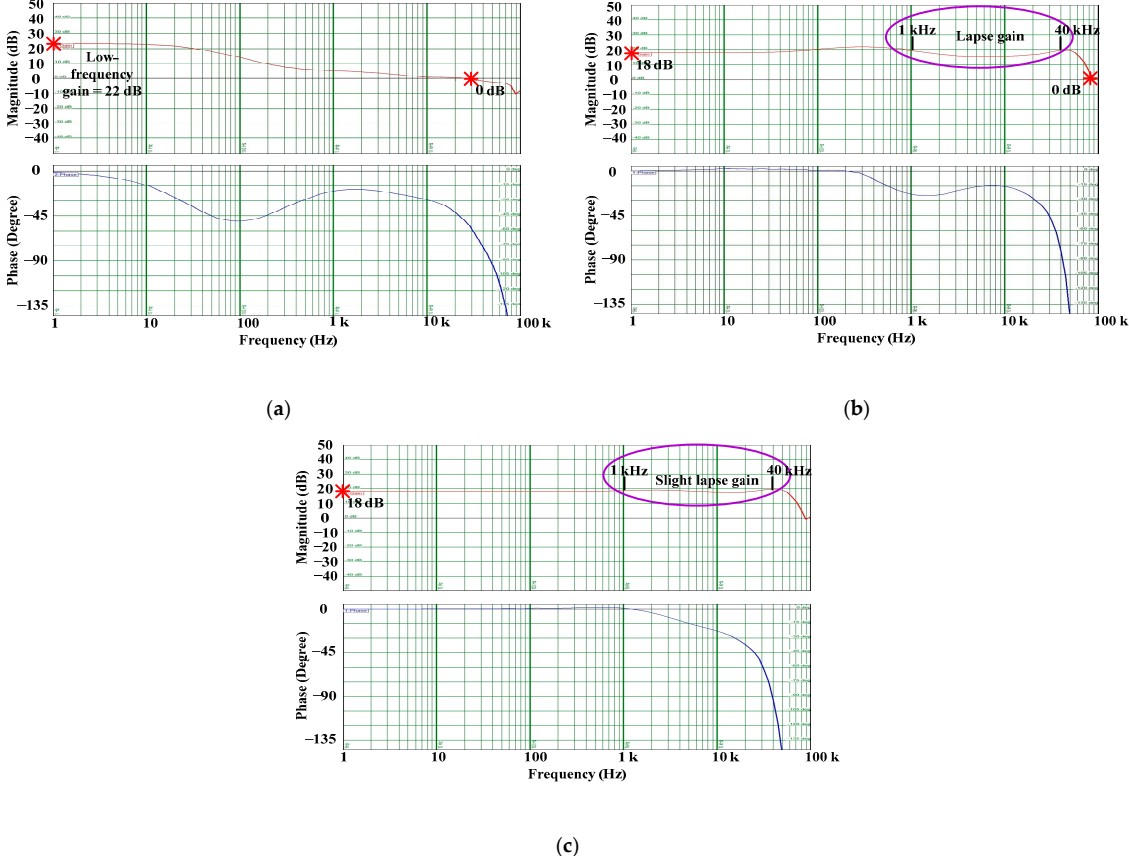

**Figure 22.** Bode plot measurements for $G_{fbv}$ and $G_{fbc}$: (**a**) $G_{fbv}$ using PI control. (**b**) $G_{fbc}$ using PI control. (**c**) $G_{fbc}$ using the ZBFS method.

Frequency response measurement of closed-loop charge current: As shown in Figure 20, with the $S_2$ turned on and the $S_1$ turned off, the FRA produced $\tilde{v}_{cfci}$, which was input to $A_2$; the $\tilde{i}_{bat}$ could then be measured by the FRA, and the Bode plot of $G_{fbc}$ was displayed in Figure 22b. The low-frequency gain was 18 dB (simulation: 18.6 dB). At the gain 0 dB, the frequency was 80 kHz (simulation: 50 kHz). The gain exhibited a lapse from 1 to 40 kHz, which could influence the system response speed. Therefore, the $f_{zc}$ should be adjusted to 5 kHz to heave a lapse gain; the subsequent Bode plot is displayed in Figure 22c.

Waveform of start-up current using PI control and ZBFS: Figure 23, Figure 24 , Figures 25–27 present the waveforms of the charge voltage $V_{bat}$ and current $I_{bat}$ during the charge start-up phase. At the charge rate of 0.5 C and SOC level of 30% (Figure 23a), the LBP started to charge from a $V_{bat}$ level of 26.92 V; the target value of average charge current was maintained at 17.5 A (0.5C). During the start-up phase, a large current overshoot occurred because the zero break frequency $f_{zc}$ was set to 49 Hz. The current overshoot peak was 34 A, which exceeded the operating CC of 16.5 A (i.e., 34 − 17.5). As illustrated in Figure 23b, the time scale of the waveform window was curtailed to 10 ms/division (ms/div.), and the following several circumstances were observed:

(1) A prolonged start-up time of 30 ms was observed before the LBPRC operated in the CC charge mode.

(2) A low-frequency current ripple of 60 Hz was observed (Figure 23); this was because an AC power source served as the input source of the LBPRC, with a power factor correction providing a DC voltage of 400 V mixing the low-frequency ripple to the PSFB-PCDR. However, the low-frequency current ripple did not damage the LBP in the charge state.

(3) The average charge current was 17.5 A, and the current overshoot peak was 34 A.

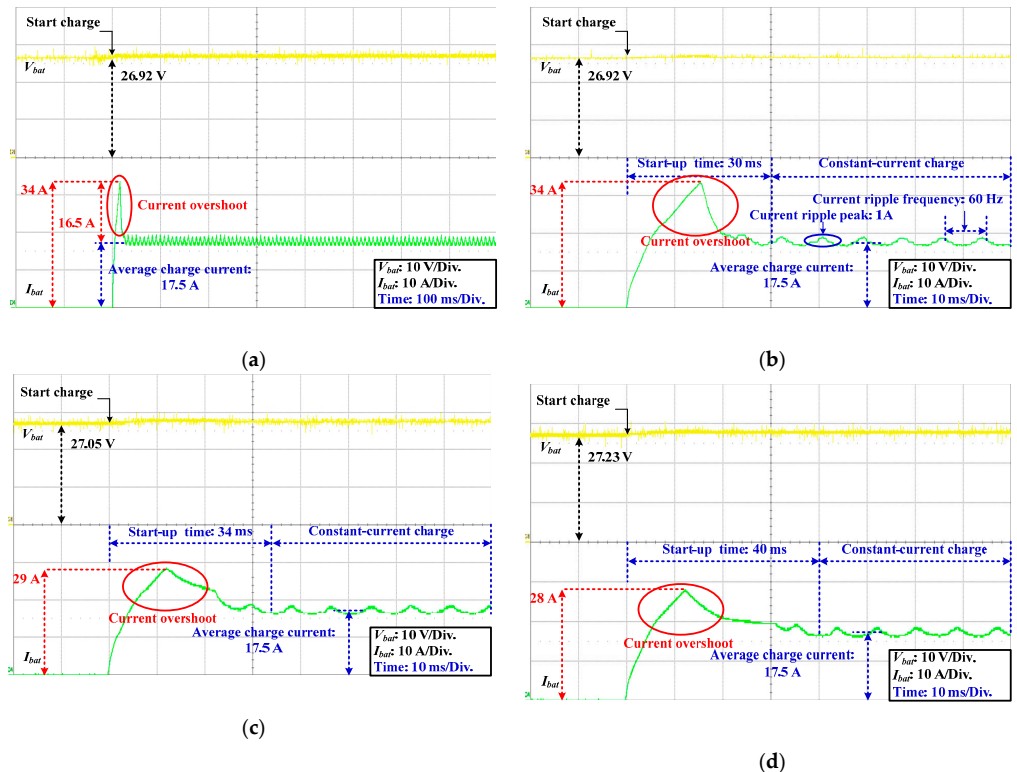

**Figure 23.** Using PI control and setting $f_{zc}$ = 49 Hz. (**a**) Start-charge voltage: 26.92 V, SOC: 30%, time scale: 100 ms/div. (**b**) Start-charge voltage: 26.92 V, SOC: 30%, time scale: 10 ms/div. (**c**) Start-charge voltage: 27.05 V, SOC: 50%, time scale: 10 ms/div. (**d**) Start-charge voltage: 27.23 V, SOC: 70%, time scale: 10 ms/div.

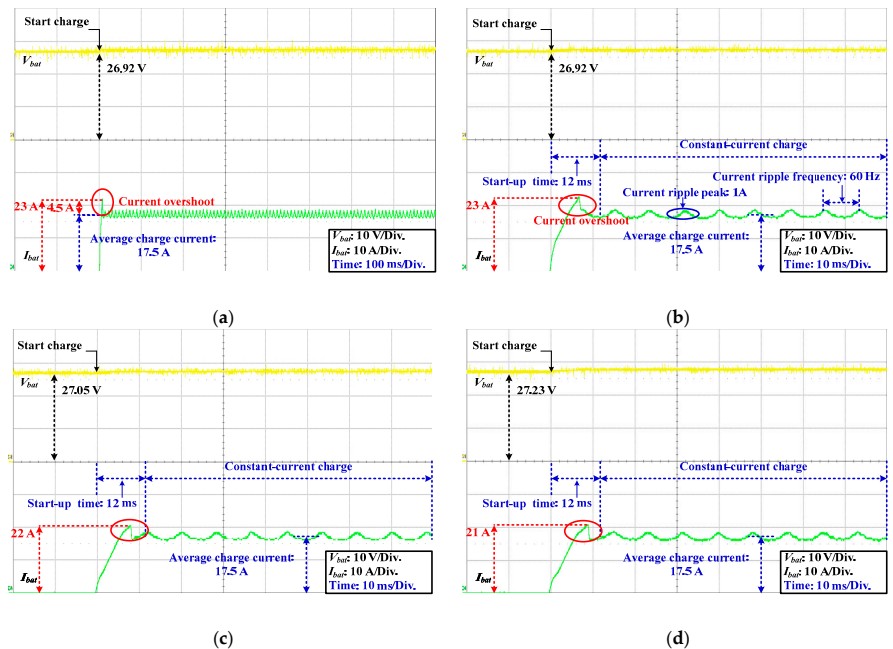

**Figure 24.** Using the ZBFS method and setting $f_{zc}$ = 5 kHz. (**a**) Start-charge voltage: 26.92 V, SOC: 30%, time scale: 100 ms/div. (**b**) Start-charge voltage: 26.92 V, SOC: 30%, time scale: 10 ms/div. (**c**) Start-charge voltage: 27.05 V, SOC: 50%, time scale: 10 ms/div. (**d**) Start-charge voltage: 27.23 V, SOC: 70%, time scale: 10 ms/div.

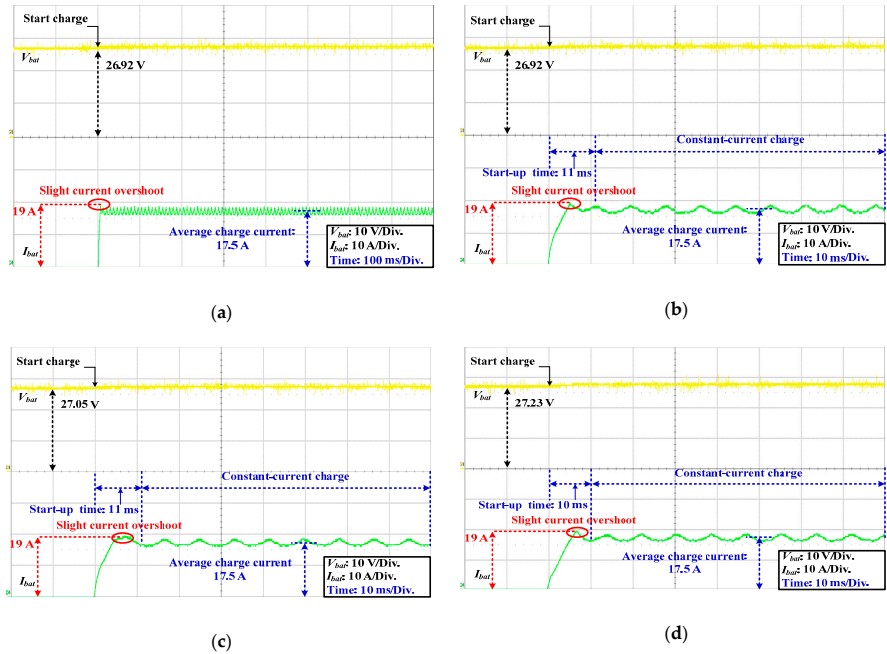

**Figure 25.** Using PSPI control. (**a**) Start-charge voltage: 26.92 V, SOC: 30%, time scale: 100 ms/div. (**b**) Start-charge voltage: 26.92 V, SOC: 30%, time scale: 10 ms/div. (**c**) Start-charge voltage: 27.05 V, SOC: 50%, time scale: 10 ms/div. (**d**) Start-charge voltage: 27.23 V, SOC: 70%, time scale: 10 ms/div.

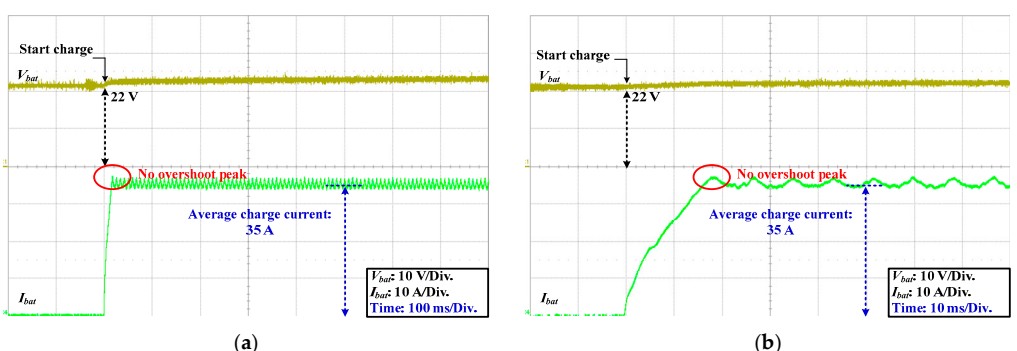

**Figure 26.** Waveforms of 1C charge using PSPI control. (**a**) Start-charge voltage: 22 V, time scale: 100 ms/div. (**b**) Start-charge voltage: 22 V, time scale: 10 ms/div.

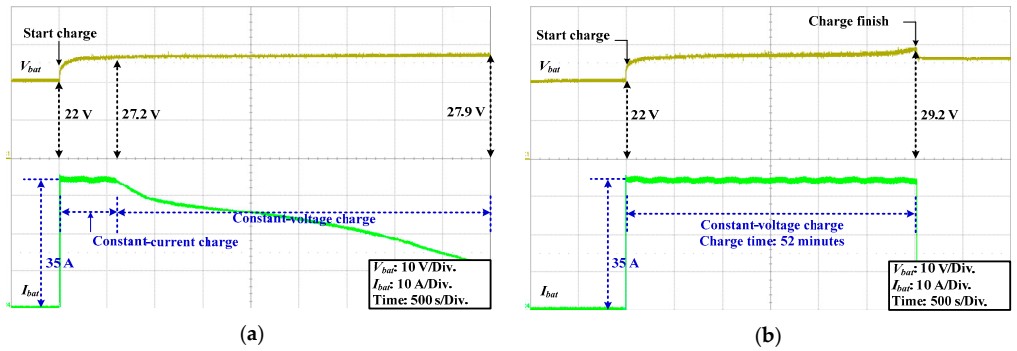

**Figure 27.** Charge profiles. (**a**) CC–CV charge strategy. (**b**) CC charge strategy.

As illustrated in Figure 23c, when the charge start voltage was 27.05 V (SOC 50%), the start-up time was 34 ms, average charge current was 17.5 A, and current overshoot peak was 29 A. As shown in Figure 23d, when the charge start voltage was 27.23 V (SOC 70%), the start-up time was 40 ms, average charge current was 17.5 A, and current overshoot peak was 28 A.

However, using the ZBFS method, the $f_{zc}$ of CFC can be shifted to 5 kHz to accelerate the response speed of the charge current loop, and the waveforms of the charge voltage and current are presented in Figure 24. In Figure 24a,b, the time scales of the waveform windows were 100 and 10 ms/div., respectively. The LBP started to charge from a $V_{bat}$ level of 26.92 V (SOC 30%); the current overshoot peak could be reduced to 23 A, the start-up time was cut to 12 ms, which was lower than in Figure 23b. Notably, the current ripple peak and frequency were 1 A and 60 Hz, respectively; the low-frequency ripple was the same as that in Figure 23b. Hence, applying the proposed ZBFS method to the LBPRC did not influence system stability. As illustrated in Figure 24c, when the charge start voltage was 27.05 V (SOC 50%), the start-up time was 12 ms, average charge current was 17.5 A, and current overshoot peak was 22 A. As illustrated in Figure 24d, when the charge start voltage was 27.23 V (SOC 70%), the start-up time was 12 ms, average charge current was 17.5 A, and current overshoot peak was 21 A. Therefore, the $f_{zc}$ shifting effectively reduces the charge current overshoot, and does not cause system instability.

Waveform of current overshoot mitigation using PSPI control: The start-charge voltage and current when the LBPRC incorporates the CFC with the PSPI control are presented in Figure 25. As illustrated in Figure 25a,b, the time scale of the waveform window was 100 and 10 ms/div., respectively. The LBP started to charge from a $V_{bat}$ level of 26.92 V (SOC 30%), the current overshoot peak was reduced to 19 A, and the start-up time was cut to 11 ms; they were lower than those in Figures 23 and 24. As illustrated in Figure 25c, when the charge start voltage was 27.05 V (SOC 50%), the start-up time was 11 ms, average charge current was 17.5 A, and current overshoot peak was 19 A. As illustrated in Figure 25d, when the charge start voltage was 27.23 V (SOC 70%), the start-up time was 10 ms, average charge current was 17.5 A, and current overshoot peak was 19 A. Compared with the PI control, ZBFS method, and PSPI control, the control method incorporating the PSPI control technology into the CFC was determined to be more effective since the current overshoot peak could be reduced to 19 A, thereby approximating the average charge current of 17.5 A; moreover, the start-up time could be curtailed to 10 ms.

Waveform of 1C charge using PSPI control: At the charge rate of 1C (CC of 35 A) in Figure 26a,b, the time scales of the waveform windows were 100 and 10 ms/div., respectively. The charge start voltage was the minimum cut-off voltage 22 V, no current overshoot was observed, and the average charge current was the CC 35 A; therefore, the proposed PSPI control was also effective for the charge rate of 1C.

CV and CC charge profiles: Figure 27 presents the LBP charge profile using the developed LBPRC. Figure 27a presented the CC–CV charge strategy; during the CC 35 A charge, the battery voltage increased gradually from 22 to 27.2 V; then, LBPRC took over as the CV output mode to charge the LBP, and the charge current gradually decreased. This experiment demonstrated that LBPRC could implement the CV charge when the LBP SOC was increased from low (20%) to high.

In Figure 27b, the single CC 35 A charges the LBP, hence the LBP voltage increased gradually from 22 to 29.2 V; the total charge time was about 52 min. This experiment demonstrated that the LBP could be charged from a low SOC of 20% to a high LBP of 83% (LBP voltage is 29.2 V) using the CC charge strategy. Both experiments confirmed that the LBPRC design was implemented using the VFC and CFC, both CV and CC charge functions could replenish the LBP. According to Figure 27b, the maximum output power of the LBPRC was 1022 W (i.e., 29.2 V × 35 A) and the charge current was 35 A (1C); therefore, the LBPRC rapid charge could be achieved for the LBP.

Moreover, the power rating of LBPRC was approximately 1 kW, and its charge current at the 1C rate was 35 A. Thus, the proposed PSFB-PCDR topology has several benefits for LBPRC applications, including zero-voltage switching, high conversion efficiency, and low-frequency current ripple reduction. The PSFB-PCDR is a commonly used topology for high-power DC–DC converter applications; the circuit component cost can be reduced in mass production to make it cost-effective.

LBPRC conversion efficiency: In this study, the power stage of the LBPRC was composed of a PFC and a PSFB-PCDR. To measure the LBPRC conversion efficiency, the LBPRC inlet inputted

a single-phase AC power source, whose output voltage and frequency were 230 $V_{rms}$ and 60 Hz, respectively. Figure 28 reports the LBPRC conversion efficiencies with different loads; the maximum efficiency was 88.8% when the output power was at 525 W; the minimum efficiency was 86.1% at 315 W. When the LBPRC operated at the maximum output power of 1050 W, the conversion efficiency was 86.7%.

In Figure 8, the maximum efficiency of the LBPRC, 88.8%, was obtained when PFC and PSFB-PCDR efficiencies were 95% and 93%, respectively. Therefore, the maximum LBPRC efficiency could be calculated as 88.8% (95% × 93% = 88%). Under this operating condition, power switches ($Q_a$ to $Q_d$) were operated in zero-voltage switching, hence the single-stage PSFB-PCDR efficiency can reach 93%; however, because of the PSFB-PCDR secondary side used diode rectification, power losses of the diodes were high. The synchronous rectification can replace the diode rectification to promote LBPRC conversion efficiency.

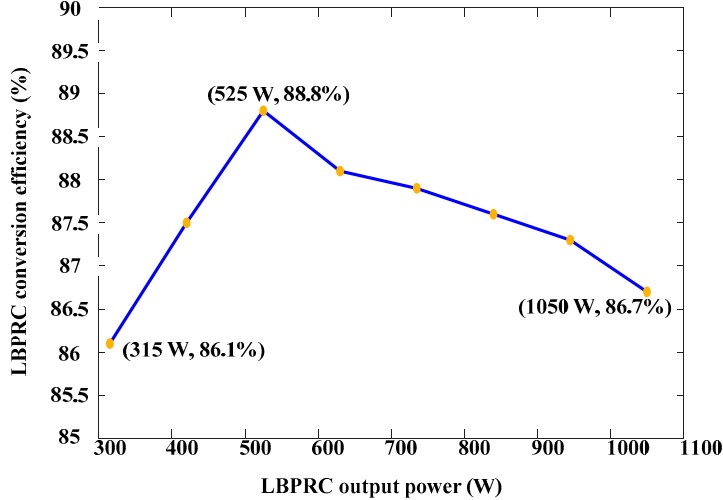

**Figure 28.** LBPRC conversion efficiency.

Comparison of performance index: Figure 29 illustrates five performance indexes for three CFC control methods. The PSPI control method achieved excellent performance indices, including the start-up current overshoot reducing, a short setting time, the small steady-state errors, a short rising time, and a short start-up time. For the ZBFS control method, four performance indices were between those observed for the PSPI and PI control methods. Finally, the PI control method integrated with the CFC was the poorest combination because four performance indices were extremely poor. Moreover, because the three control methods can implement the PI control during the steady-state operation, they have small steady-state errors.

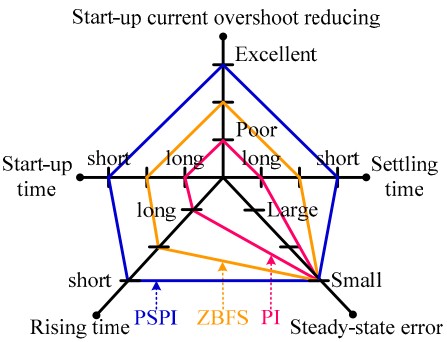

**Figure 29.** Performance indices for PSPI control, the ZBFS method, and PI control.

LBPRC prototype: Figure 30 is a photo of the practical charge system. The input power of the LBPRC prototype comes from the AC source, and its output side connects to the LBP. To analyze the frequency response, the FRA is required and the Bode plot measurements can be displayed on the computer monitor. In addition, the experimental waveforms can be measured and displayed by the oscilloscope.

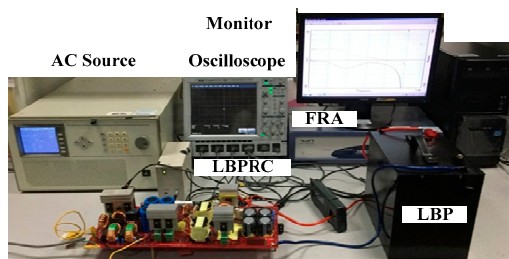

**Figure 30.** A photo of the practical charge and measurement system.

## 7. Conclusions

This study presented the development and implementation of an LBPRC for the charge of LiFePO$_4$ battery packs. The focus was on frequency response analyses of charge voltage and current; therefore, the TTS and PNGV battery models were applied to derive small-signal transfer functions. Consequently, the LBPRC could utilize both CV and CC power to replenish the LBP. This study makes the following contributions:

(1) This study is the first to propose combinations of TTS and PNGV battery models in an LBPRC application alongside ZBFS and PSPI controls to mitigate the problem of current overshooting.

(2) The proposed equivalent circuit, which incorporates the TTS and PNGV battery models with the wire resistance-inductance of the power cable, is a reformatory model for the small-signal analysis in the LBPRC application.

(3) On the basis of the TTS and PNGV battery models, high-order transfer functions were derived for the charge voltage and current loops. Although establishing the mathematical model was a complex and time-consuming process, the frequency response simulations could approximate the practical system. Therefore, the loop compensation could follow the proposed design procedure to achieve the zero-pole frequency setting, low-frequency gain heaving, and phase margin enhancement.

(4) Few studies have discussed CC control designs or how to mitigate charge current overshoot. According to the Bode plot of the feedback current loop, the CFC compensation uses ZBFS; therefore, the current overshoot can be effectively reduced in the charge start-up phase.

(5) The study also presents a new control method, namely PSPI control, integrated into a CFC to mitigate the problem of charge current overshoot during the LBPRC start-up phase. The benefit of this method is that no compensative element needs to be readjusted or redesigned, hence the prototypal PI parameter can be applied to achieve the PSPI control.

(6) Three measurement methods, including the electrical parameter of PNGV battery model, the open-loop frequency response of LBPRC, and the closed-loop frequency response of LBPRC, were presented and discussed in this study. These methods are prerequisite designing processes to develop LBPRCs. Finally, an LBPRC prototype and practical charge system are also presented in this paper.

**Funding:** This research was funded by [the Ministry of Science and Technology, Taiwan (R.O.C.)]. The grant number: [MOST 104–2218–E–236–002], [MOST 105–2221–E–236–003], and [MOST 107–2221–E–131–008].

**Acknowledgments:** Author acknowledges the Ministry of Science and Technology, Taiwan (R.O.C.) supplying a research fund.

**Conflicts of Interest:** The authors declare no conflict of interest.

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
