# Peer review of "A Reformatory Model Incorporating PNGV Battery and Three-Terminal-Switch Models to Design and Implement Feedback Compensations of LiFePO4 Battery Chargers"

_electronics, doi:10.3390/electronics8020126_

Round 1

Reviewer 1 Report

The manuscript is interesting and well written.

On row 59 “boot converter” should be “boost converter.”

What value is the primary side capacitor Cp (row 145)?

The maximum efficiency of 88.8% (row 582) is rather low for that class converters, which can compromise your design. Have you achieved zero voltage switching on the primary side and zero current switching on the secondary side?

Author Response

The manuscript is interesting and well written.

Response: Thank you for considering my manuscript and for your positive evaluation.

On row 59 “boot converter” should be “boost converter.”

Response: Please review the row 61 in page 2. The error has modified in the new manuscript.

What value is the primary side capacitor Cp (row 145)?

Response: Please review the row 158 in page 6. Cb represents the blocking capacitance in this study. The capacitor value was 2.2 mF (Table 7).

The maximum efficiency of 88.8% (row 582) is rather low for that class converters, which can compromise your design. Have you achieved zero voltage switching on the primary side and zero current switching on the secondary side?

Response: Please review the row 585 in page 25. In this study, the power stage of the LBPRC was composed of a PFC and a PSFB-PCDR. To measure the LBPRC conversion efficiency, the LBPRC inlet inputted a single-phase AC power source, whose output voltage and frequency were 230 Vrms and 60 Hz, respectively. Figure 28 recorded LBPRC conversion efficiencies in different loads; the maximum efficiency was 88.8%, when the output power was at 525 W; the minimum efficiency was 86.1% at 315 W. When the LBPRC operated at the maximum output power 1050 W, the conversion efficiency was 86.7%.

In Figure 8, the maximum efficiency of the LBPRC, 88.8%, was obtained when PFC and PSFB-PCDR efficiencies were 95% and 93%, respectively. Therefore, the maximum LBPRC efficiency could be calculated as 88.8% (95% × 93% = 88%). Under this operating condition, power switches (Qa to Qd) were operated in zero-voltage switching, hence the single-stage PSFB-PCDR efficiency could achieve 93%; however, because of the PSFB-PCDR secondary side used the diode rectification, power losses on diodes were high. The synchronous rectification can replace with the diode rectification to promote the LBPRC conversion efficiency.

Reviewer 2 Report

see attached file

Author Response

Many thanks for you to give me an opportunity to revise my manuscript. Please see the attachment.

This manuscript is a resubmission of an earlier submission. The following is a list of the peer review reports and author responses from that submission.

Round 1

Reviewer 1 Report

The manuscript is interesting and well written.

On row 59 “boot converter” should be “boost converter.”

What value is the primary side capacitor Cp (row 145)?

The maximum efficiency of 88.8% (row 582) is rather low for that class converters, which can compromise your design. Have you achieved zero voltage switching on the primary side and zero current switching on the secondary side?

Author Response

(The authors gave the same response as above.)

Reviewer 2 Report

my comments in the attached file

Author Response

Because of responses include images. Please review the attachment.
